# Comparative analyses of Netherton syndrome patients and *Spink5* conditional knock-out mice uncover disease-relevant pathways

Evgeniya Petrova [1✉], Jesús María López-Gay[2,3], Matthias Fahrner [4], Florent Leturcq [1], Jean-Pierre de Villartay [5], Claire Barbieux [1], Patrick Gonschorek [6], Lam C. Tsoi[7,8,9], Johann E. Gudjonsson [7], Oliver Schilling[4] & Alain Hovnanian [1,10,11✉]

Netherton syndrome (NS) is a rare skin disease caused by loss-of-function mutations in the serine peptidase inhibitor Kazal type 5 (*SPINK5*) gene. Disease severity and the lack of efficacious treatments call for a better understanding of NS mechanisms. Here we describe a novel and viable, *Spink5* conditional knock-out (cKO) mouse model, allowing to study NS progression. By combining transcriptomics and proteomics, we determine a disease molecular profile common to mouse models and NS patients. *Spink5* cKO mice and NS patients share skin barrier and inflammation signatures defined by up-regulation and increased activity of proteases, IL-17, IL-36, and IL-20 family cytokine signaling. Systemic inflammation in *Spink5* cKO mice correlates with disease severity and is associated with thymic atrophy and enlargement of lymph nodes and spleen. This systemic inflammation phenotype is marked by neutrophils and IL-17/IL-22 signaling, does not involve primary T cell immunodeficiency and is independent of bacterial infection. By comparing skin transcriptomes and proteomes, we uncover several putative substrates of tissue kallikrein-related proteases (KLKs), demonstrating that KLKs can proteolytically regulate IL-36 pro-inflammatory cytokines. Our study thus provides a conserved molecular framework for NS and reveals a KLK/IL-36 signaling axis, adding new insights into the disease mechanisms and therapeutic targets.

[1] INSERM UMR 1163, Laboratory of Genetic Skin Diseases, Imagine Institute and University of Paris, Paris, France. [2] Institut Curie, PSL Research University, CNRS UMR 3215, INSERM U934, Paris F-75248 Cedex 05, France. [3] Sorbonne University, UPMC University Paris 06, CNRS, CNRS UMR 3215, INSERM U934, F-75005 Paris, France. [4] Institute for Surgical Pathology, Medical Center, Faculty of Medicine, University of Freiburg, Germany; German Cancer Consortium (DKTK) and Cancer Research Center (DKFZ), Freiburg, Germany. [5] Imagine Institute, Laboratory "Genome Dynamics in the Immune System", INSERM UMR 11635 Paris, France. [6] Institute of Chemical Sciences and Engineering, School of Basic Sciences, Ecole Polytechnique Fédérale de Lausanne (EPFL), Lausanne CH-1015, Switzerland. [7] Department of Dermatology, University of Michigan Medical School, Ann Arbor, MI, USA. [8] Department of Computational Medicine & Bioinformatics, University of Michigan Medical School, Ann Arbor, MI, USA. [9] Department of Biostatistics, School of Public Health, University of Michigan, Ann Arbor, MI, USA. [10] Department of Genomic Medicine of rare diseases, Necker Hospital for Sick Children, Assistance Publique des Hôpitaux de Paris (AP-HP), Paris, France. [11] University of Paris Cité, Paris, France. ✉email: evgeniya.petrova@inserm.fr; alain.hovnanian@inserm.fr

Netherton syndrome (NS) is a severe autosomal recessive skin disease caused by loss-of-function mutations in the serine protease inhibitor Kazal-type 5 (SPINK5) gene, leading to deficiency of its protein product lymphoepithelial Kazal-type-related protease inhibitor (LEKTI)[1]. The disease is characterized by generalized erythroderma, scaling, hair shaft defect and atopy. NS affects 1:200 000 persons worldwide and can be life-threatening in newborns and infants. Loss of skin barrier function in NS leads to severe dehydration, disruption of the skin microbiome homeostasis and severe, chronic or relapsing skin inflammation most often accompanied by allergic reactions[2,3]. However, the contribution of immune system defects to recurrent infections seen in NS patients remains debated[4]. Currently, there are no effective therapies for NS that target the disease pathophysiology and treatment is mostly symptomatic.

LEKTI is a protease inhibitor that is expressed in the most differentiated viable layers of stratified epithelia including skin and esophagus and in the thymic medullary epithelium[5]. In the epidermis, LEKTI is expressed and secreted by keratinocytes at the interface between the granular and cornified layers[5,6]. Serine proteases, such as kallikrein-related peptidases (KLK), are also produced and secreted at the same location, where they play role in desquamation, antimicrobial defense and lipid permeability[7]. In the skin, LEKTI inhibits the serine proteases KLK5, KLK6, KLK7, KLK13, KLK14 and Cathepsin G[8–12] and the cysteine protease Caspase 14[13]. The balance between LEKTI-mediated protease inhibition and proteolytic activity is an important mechanism that ensures a gradual desquamation process during the normal renewal of the epidermis[12]. Accordingly, LEKTI deficiency leads to unrestrained proteolytic activity in the epidermis, resulting in premature stratum corneum detachment and subsequent skin barrier defect[14,15]. Uncontrolled KLK activity has two major impacts on the skin: (i) skin barrier defect resulting from proteolytic degradation of (corneo)desmosomal cadherins[15,16] and stratum corneum lipid-processing enzymes[17], and (ii) skin inflammation through proteolytic activation of PAR-2 signaling[18], and proteolytic processing of complement C3, IL-1B, or antimicrobial peptides[19–21].

LEKTI is functionally conserved in mice and Spink5-deficient mouse models mimic the phenotype of NS[22]. However, currently there is incomplete knowledge of the skin and systemic inflammation molecular profiles in Spink5-deficient mice and how they compare to the molecular features of the disease in humans. Moreover, there is a need of a viable Spink5 KO mouse model of NS that is fully characterized at the molecular level and can thus aid in the pre-clinical testing of drug candidates for NS. Constitutive Spink5-deficient mice[14,23–25] reproduce the NS skin phenotype, but do not allow to study disease progression, since they die within few hours after birth. Grafting of Spink5-deficient epidermis onto wild-type mice[26] or TALEN-mediated generation of Spink5 mosaic mice[27] has permitted to study Spink5-deficient adult skin; however, these methods are technically challenging and results can be confounded with artifacts of the method used to generate the model.

Additionally, viable mouse models based on the over-expression of different human KLK genes such as KLK5[28], KLK6[29], KLK7[30], or KLK14[31] and of the murine epidermal chymotrypsin-like elastase 2a gene Cela2a[32] can reproduce certain features of NS phenotype. However, in all these models, LEKTI expression remains intact and, since LEKTI can inhibit the activity of several proteases in the skin, it is likely that the skin phenotype and the molecular profile of adult Spink5-deficient skin is different from that of KLK-overexpressing mice. Further drawbacks of KLK transgenic mice are the unstable nature of transgene expression, which can result in skin phenotype attenuation or loss, and the possible differences in endogenous substrate targets between human and mouse KLKs.

Here, we characterize a Cre/loxP-based, inducible Spink5 knock-out mouse model that is viable in adult stages and reproduces the NS phenotype. This Spink5 conditional knock-out (cKO) mouse model allowed us to perform detailed transcriptome and proteome analyses of the skin and immune system in adult mice and to compare the results to the skin molecular profile of NS patients. We thus identified shared molecular signatures of skin barrier defect and inflammation in Spink5 cKO mice and NS patients, suggesting an important role of the immune system in NS. Moreover, we uncovered a direct cross-talk between epidermal KLK activity and IL-36 pro-inflammatory cytokines, pointing out a key role of KLKs in controlling both skin barrier and skin immunity.

## Results

### Spink5 conditional knock-out mice are viable and replicate the clinical features of NS.

To generate a Spink5 conditional knock-out (cKO) mouse model, we engineered mice harboring floxed (conditional) alleles, in which Spink5 exon 3 is flanked by loxP sites (Fig. 1a and Supplementary Fig. 1). Mice homozygous for the Spink5 floxed allele (Spink5^fl/fl) or carrying one Spink5 floxed allele and one Spink5 constitutive knock-out allele[14] were crossed with KRT14-CreERT2 transgenic mice[33] to obtain tissue-specific tamoxifen-inducible KRT14-CreERT2^(Tg/0)/Spink5^fl/fl or KRT14-CreERT2^(Tg/0)/Spink5^fl/- mice. We determined that optimal induction of CreERT2 recombination activity in this mouse model occurs in young mice at 4–5 weeks of age (Fig. 1b). Tissue-specific deletion of Spink5 was confirmed by PCR-based detection of Spink5 excised/knock-out allele (Δex3) in the epidermis and other epithelial tissues, where the KRT14 promoter is active (Fig. 1c and Supplementary Fig. 20). The level of Spink5 deletion varied between individual mice and tissues (Fig. 1c, Supplementary Fig. 2a, b, and Supplementary Fig. 22). Induction of Spink5 deletion in the epidermis of Spink5 cKO mice resulted, within 10 days after starting tamoxifen administration, in the development of skin lesions, which resembled those of NS patients (Fig.1d and Supplementary Fig. 2c). Approximately 40% of both KRT14-CreERT2^(Tg/0)/Spink5^fl/fl and KRT14-CreERT2^(Tg/0)/Spink5^fl/- mice showed spontaneous floxed allele excision, which is due to the leaky nature of the KRT14-CreERT2 transgene as previously described[34,35]. NS-like skin lesions developed in about 50% of the mice with spontaneous floxed allele excision as early as 2 weeks after birth (Supplementary Fig. 2d).

Spink5 cKO mice developed red, scaly and crusty skin with alopecic areas secondary to scratching behavior (Fig. 1d and Supplementary Fig. 2c, d). This phenotype was accompanied by weight loss and signs of emaciation in the most severe cases (Supplementary Fig. 2e). The severity of skin lesions varied to a similar degree in both KRT14-CreERT2^(Tg/0)/Spink5^fl/fl and KRT14-CreERT2^(Tg/0)/Spink5^fl/- mice (Supplementary Fig. 2f). Their survival depended on the severity and extent of skin lesions with a mean survival time of 5 weeks and 8 weeks for KRT14-CreERT2/Spink5^fl/- and KRT14-CreERT2/Spink5^fl/fl mice, respectively (Supplementary Fig. 2g). Since the skin phenotypes of mice carrying either two Spink5 floxed (conditional knock-out) alleles (KRT14-CreERT2^(Tg/0)/Spink5^fl/fl) or one floxed allele and one Spink5 constitutive knock-out allele (KRT14-CreERT2^(Tg/0)/Spink5^fl/-) were mostly indistinguishable, Spink5 cKO designation will be used for both genotypes throughout the manuscript, unless otherwise specified.

We confirmed the reduction of Spink5 expression in Spink5 cKO skin at the levels of protein (Fig.1e and Supplementary Figs. 2h, j–m and 22) and mRNA (Fig. 1f and Supplementary Fig. 2i). The incomplete reduction of Spink5 expression in Spink5 cKO skin suggests that CreERT2-mediated excision of the floxed allele did

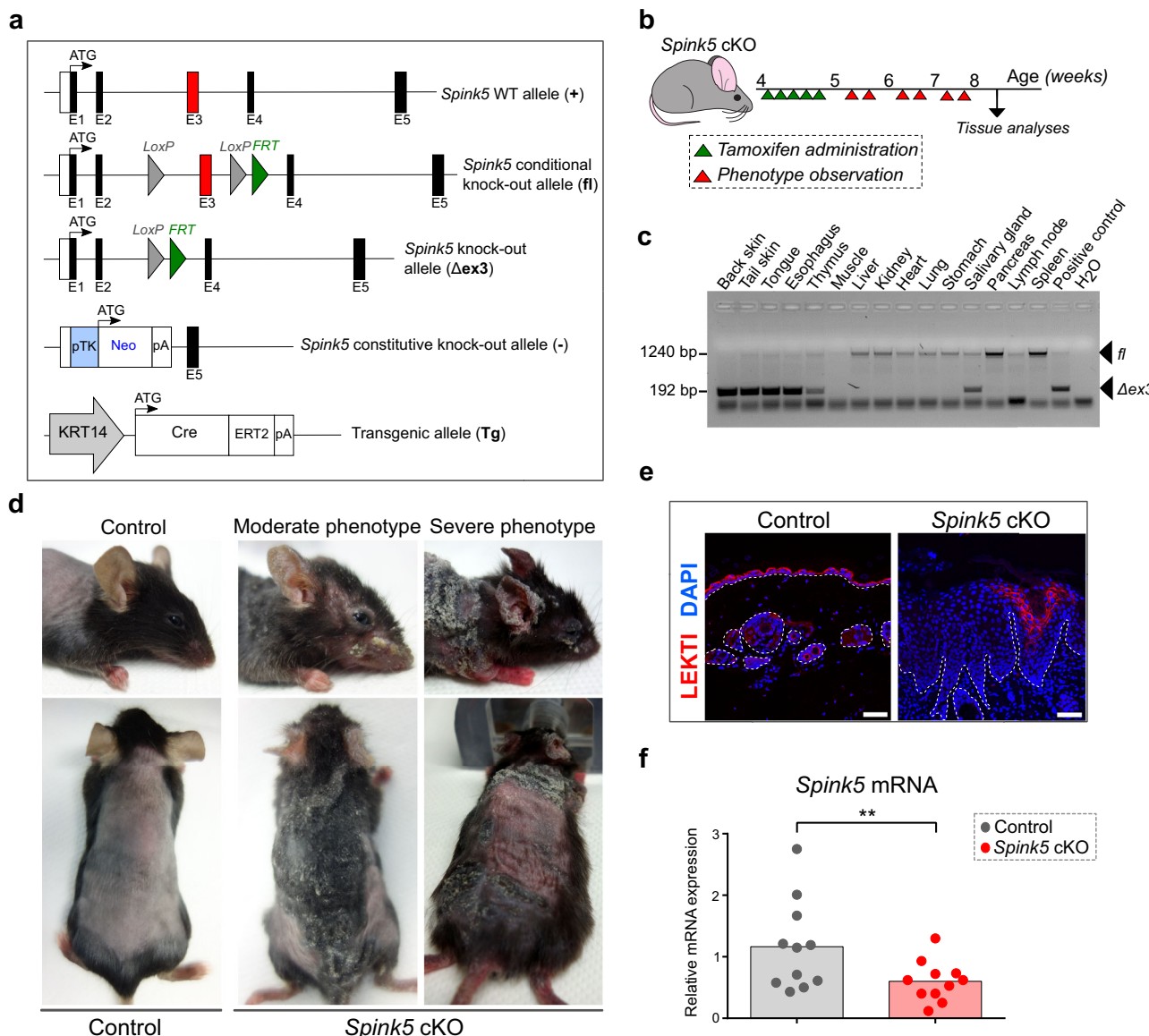

**Fig. 1 Conditional deletion of *Spink5* using KRT14-CreERT2 results in viable mice that develop skin lesions resembling Netherton syndrome.**
**a** Schematic of the *Spink5* gene-targeted locus and alleles present in *Spink5* conditional knock-out mice. The first 5 exons of the *Spink5* WT allele are shown (boxes) with targeted exon 3 marked in red. The *Spink5* conditional knock-out (floxed) allele and the knock-out (excised / Δex3) allele resulting from Cre-mediated excision of the loxP-flanked exon 3 are shown. The *Spink5* constitutive knock-out allele (-) and the transgenic allele (Tg) expressing Cre-ERT2 under the human Keratin 14 (KRT14) promoter are shown. **b** Schematic of the protocol used for tamoxifen-inducible CreERT2-mediated *Spink5* deletion in *Spink5* cKO mice. **c** Image of agarose gel electrophoresis analysis of PCR done on genomic DNA to detect CreERT2-mediated excision of exon 3 in different tissue samples collected from an individual *Spink5* conditional knock-out mouse. Arrows indicate bands corresponding to conditional (fl) and knock-out (Δex3) alleles. **d** Images of control and *Spink5* conditional knock-out (cKO) mice with moderate and severe back skin and face skin phenotypes of at the age of 8 weeks. The back skin was shaved before imaging. **e** Confocal microscopy images of LEKTI immunofluorescence staining (red) in back skin paraffin sections of control and *Spink5* cKO mice. Nuclei are counterstained with DAPI (blue). The dermal-epidermal junction is outlined with a white dashed line. Scale bars: 50 μm. **f** *Spink5* mRNA expression levels measured in back skin samples of control and *Spink5* cKO mice using RT-qPCR. Data in **c** is representative of three independent experiments. Data in **d** and **e** are representative of at least 10 and 5 independent experiments, respectively. The graph in **f** show means (bars) and scatter plot, where dots correspond to individual mice (n = 11 per group). Statistical significance was determined using two-tailed non-parametric Wilcoxon matched-pairs signed rank test: \*\**p* < 0.01. Control mice are *Spink5*^fl/fl and/or *Spink5*^fl/- ; *Spink5* cKO mice are *KRT14-CreERT2*^(Tg/0)/*Spink5*^fl/fl and/or *KRT14-CreERT2*^(Tg/0)/*Spink5*^fl/- . See also Supplementary Figs. 1 and 2.

not occur in all cells of the epidermal basal layer (Fig. 1e, f). Lesional skin showed significantly lower levels of LEKTI protein and *Spink5* mRNA expression as compared to the outwardly normal looking non-lesional skin or WT control skin (Supplementary Fig. 2h–m). Moreover, the severity score of lesional back skin biopsies correlated significantly with the decrease of *Spink5* mRNA and LEKTI protein expression (Supplementary Fig. 2n, o).

**The skin of *Spink5* cKO mice displays increased kallikrein-related protease activity, defective skin barrier and abnormal epidermal differentiation.** Previous studies in mice and NS patients have established that LEKTI deficiency leads to the unrestrained activity of epidermal serine proteases. LEKTI is an inhibitor of the epidermal trypsin-like serine proteases KLK5, KLK6, KLK13 and KLK14 and the epidermal chymotrypsin-like

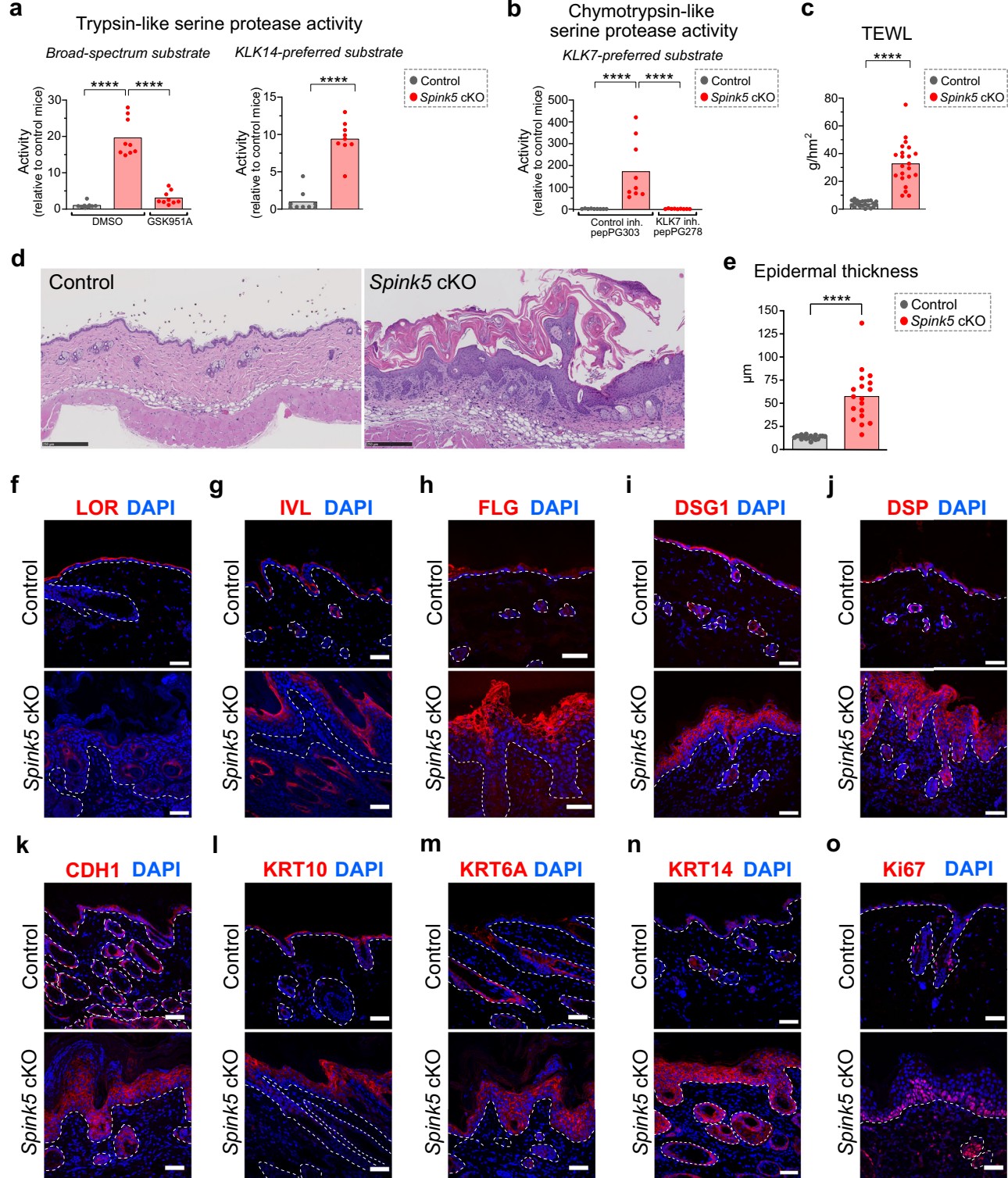

serine protease KLK7[10,11]. We measured the activity of these proteases in protein extracts from *Spink5* cKO lesional skin using fluorogenic peptide substrates. The activity of trypsin-like serine proteases in skin extracts from *Spink5* cKO mice was increased by 19-fold as compared to control mice (Fig. 2a). To estimate the relative contribution of KLK5 and KLK14 to this trypsin-like serine protease activity in *Spink5* cKO skin, we measured Boc-VPR-amc substrate cleavage in skin extracts co-incubated with the KLK5-specific inhibitor GSK951A[36], or we used the KLK14-preferred peptide substrate Ac-WAVR-amc[37]. Addition of

GSK951A reduced the fluorescence signal by 85% (Fig. 2a). On the other hand, the activity of KLK14 in *Spink5* cKO skin was increased by 9-fold relative to control mice (Fig. 2a), which represents approximately 50% of the increase in trypsin-like protease activity measured with the broad-spectrum substrate Boc-VPR-amc. Thus, these results suggest that KLK5 and KLK14 are the major contributors to the observed increase in trypsin-like serine protease activity. Regarding chymotrypsin-like serine proteases, the activity of KLK7, as measured with the KLK7-preferred substrate KHLY-amc[38], was highly increased (by 172-

**Fig. 2 The skin of *Spink5* conditional KO mice displays increased serine protease activity, skin barrier defect and abnormal epidermal differentiation.**
**a** Measurements of trypsin-like serine protease activity in protein extracts from back skin of control (gray) and *Spink5* cKO (red) mice using the broad-spectrum fluorogenic substrate for trypsin-like serine proteases Boc-VPR-amc (left panel) and the KLK14-preferred fluorogenic peptide substrate Ac-WAVR-amc (right panel). **b** Measurements of chymotrypsin-like serine protease activity in protein extracts from back skin of control mice (gray) and *Spink5* cKO mice (red) using the KLK7-preferred fluorogenic peptide substrate KHLY-amc. In **a** and **b**, activity is expressed as a ratio of the fluorescence intensity value measured in each skin sample to the mean of the fluorescence intensity values measured in skin extracts from control mice. The addition of the KLK5-specific inhibitor GSK951A or the KLK7-specific inhibitor pepPG278 serves as control to estimate the percent of trypsin-like or chymotrypsin-like protease activities due to KLK5 or KLK7 activation, respectively. As negative controls for protease activity inhibition, DMSO or the non-specific inhibitor pepPG303 were added. **c** Transepidermal water loss (TEWL) measurements performed on the back skin of control mice (gray) and lesional back skin of *Spink5* cKO mice (red). **d** Hematoxylin and eosin staining of back skin paraffin sections from control mice (left panel) and *Spink5* cKO littermate mice (right panel) with severe phenotype. Scale bars: 250 μm. **e** Quantification of epidermal thickness performed on images of hematoxylin- and eosin-stained back skin paraffin sections from control (gray) and *Spink5* cKO (red) littermate mice. **f–o** Immunofluorescence staining (red) of Loricrin (LOR, **f**), Involucrin (IVL, **g**), Filaggrin (FLG, **h**), Desmoglein1 (DSG1, **i**), Desmoplakin (DSP, **j**), E-cadherin (CDH1, **k**), Keratin10 (KRT10, **l**), Keratin 6A (KRT6A, **m**), Keratin 14 (KRT14, **n**) and the marker of proliferation Ki-67 (**o**) in back skin sections from control (upper panel) and *Spink5* cKO (lower panel) littermate mice. Nuclei are counterstained with DAPI (blue). Scale bars: 50 μm. The dermal-epidermal junction is outlined with a white dashed line. Images are representative of immunofluorescence staining performed on samples from at least five different mice. The graphs in **a–c** and **e** show means (bars) and scatter plots, where data points correspond to individual mice ($n \geq 8$ per group). Each dot represents the mean of all measurements for an individual mouse. Statistical significance was determined using a two-tailed non-parametric Mann-Whitney test: ****$p < 0.0001$. Control mice are Spink5$^{fl/fl}$ and/or Spink5$^{fl/-}$; *Spink5* cKO mice are *KRT14-CreERT2$^{(Tg/0)}$/Spink5$^{fl/fl}$* and/or *KRT14-CreERT2$^{(Tg/0)}$/Spink5$^{fl/-}$*. See also Supplementary Figs. 3 and 4.

fold) in lesional skin of *Spink5* cKO mice relative to controls (Fig. 2b). Co-incubation of the skin extracts with the KLK7-specific inhibitor pepPG278[39] completely abolished the fluorescence signal, suggesting that KLK7 is the main active chymotrypsin-like serine protease in the skin of *Spink5* cKO mice. Using the same assay, we compared the skin protease activity of *Spink5* cKO mice to that of Tg.*hKLK5* mice - a previously described viable mouse model of NS[28]. Tg.*hKLK5* mice develop a severe skin barrier defect due to hKLK5 over-expression and activity, despite intact LEKTI expression. In the skin of Tg.*hKLK5* mice, the activities of trypsin-like serine proteases, KLK14 and KLK7 were increased by 13-fold, 16-fold, and 28-fold, respectively (Supplementary Fig. 3a, b).

Consistent with the observed hyperactivity of tissue kallikrein-related peptidases, the epidermal barrier in *Spink5* cKO mice was severely compromised. The average of transepidermal water loss (TEWL) values measured on dorsal skin was 8 times higher in *Spink5* cKO mice than in control mice (Fig. 2c). The TEWL values in *Spink5* cKO mice were maintained high over time (Supplementary Fig. 4a) and their magnitude correlated strongly with the severity of skin lesions (Supplementary Fig. 4b).

Histological analyses of skin samples from *Spink5* cKO mice revealed stratum corneum detachment with parakeratosis and hyperkeratosis (Fig. 2d and Supplementary Fig. 4d, e). Other prominent features of *Spink5* cKO skin were epidermal thickening and the presence of subcorneal neutrophilic microabscesses (Fig. 2d and Supplementary Fig. 4d, e). The epidermal thickness of *Spink5* cKO skin was significantly increased (by 4-fold) compared to control mice (Fig. 2e) and correlated significantly with the severity score of skin lesions (Supplementary Fig. 4c). As observed previously in NS patients[4] and other mouse models of NS[14,31], *Spink5* cKO mice also displayed hair loss, including loss of vibrissae hairs (whiskers), at sites of skin lesions (Fig. 1d and Supplementary Fig. 4f, g).

To assess how deletion of *Spink5* in adult mice and the resulting increase in epidermal serine protease activity affect epidermal differentiation, we analyzed by immunofluorescence microscopy the expression pattern and level of several markers of epidermal differentiation and proliferation (Fig. 2f–o and Supplementary Fig. 4h–m). We did not observe significant changes in the overall expression level for most of the markers analyzed, except for the expression of Loricrin (LOR), Keratin 14 (KRT14), Keratin 6A (KRT6A) and Ki67. LOR was absent in *Spink5* cKO skin (Fig. 2f) consistent with previous findings in NS

patient skin[3]. KRT6A was expressed in lesional interfollicular skin in *Spink5* cKO mice (Fig. 2m), consistent with the known upregulation of KRT6A expression in interfollicular keratinocytes upon trauma, wounding or hyperproliferative skin diseases such as psoriasis. The expression of the basal layer marker KRT14 was strongly increased in all epidermal layers, suggesting abnormal keratinocyte proliferation and differentiation (Fig. 2n and Supplementary Fig. 4m–m′). Accordingly, the expression of the cell proliferation marker Ki67 was increased in all basal layer cells as well as in some suprabasal layer cells in *Spink5* cKO epidermis (Fig. 2o).

Altogether, the above findings confirm the similarity of skin lesions in *Spink5* cKO mice and NS patients and further indicate that *Spink5* cKO mice display increased skin serine protease activity predominated by KLK7 over-activation, resulting in skin barrier defect and abnormal epidermal differentiation and proliferation.

**Transcriptome profiling of *Spink5* cKO skin reveals features of acute and chronic skin inflammation**. To better understand how the *Spink5* cKO mouse model compares with NS patients, we further analyzed the molecular features of lesional skin in *Spink5* cKO mice. We performed bulk RNA sequencing (RNAseq) of *Spink5* cKO lesional back skin and control mouse back skin and then compared the *Spink5* cKO differential gene expression profile to that of NS patient lesional skin[3] (Supplementary Fig. 5a, d). We found a significant positive correlation between the global differential gene expression profile in *Spink5* cKO lesional skin and NS patient lesional skin (Fig. 3a). In parallel, we extended these comparative skin transcriptome analyses to the previously described mouse models of NS - the viable Tg.*hKLK5* mice[28] and *Spink5$^{-/-}$* mice, which display early postnatal lethality[14] (Fig. 3b, c and Supplementary Fig. 5b, c). Statistical analyses comparing the correlations among the different mouse models of NS revealed that the differential gene expression profile of *Spink5* cKO skin is significantly the most similar to that of NS patient lesional skin (Fig. 3a–c and Supplementary Fig. 5g). These results suggest that at the skin transcriptome level *Spink5* cKO mice are the best model mimicking NS patient lesional skin.

Gene ontology (GO) enrichment analyses of the up-regulated and down-regulated differentially expressed genes (DEGs) shared between *Spink5* cKO and NS patient skin (Fig. 3d, e) resulted in 85 significantly enriched biological process GOs. To facilitate the

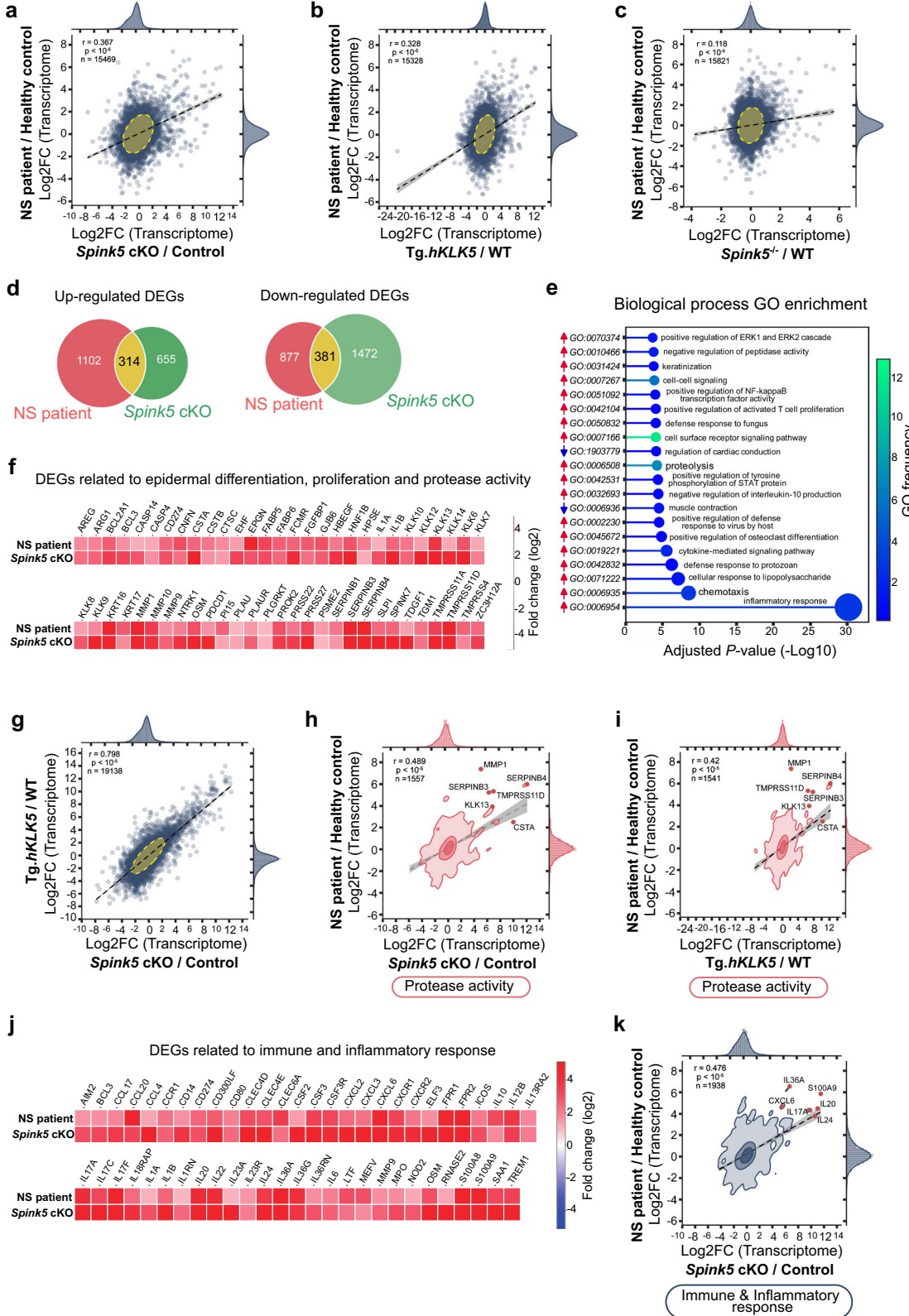

study of these numerous GO terms, we grouped them into 8 major groups depending on the type of biological process (Supplementary Figs. 6a and 7a, b and Supplementary Data 1). In agreement with our previous tissue-level analyses of *Spink5* cKO skin (Fig. 2) and NS patients' skin[3], we found significant enrichment of biological process GO terms related to keratinization, cell proliferation and protease activity. Interestingly,

comparative analyses done only with the set of genes annotated in the enriched GOs related to protease activity resulted in an overall better correlation as compared to the global gene expression correlations (Fig. 3h and Supplementary Data 2), indicating the key role of protease activity in NS pathogenesis. The biological process GOs related to protease activity were among the 20 most significantly enriched ones and were only

**Fig. 3 Transcriptome profiling of *Spink5* cKO skin and comparison to the lesional skin transcriptomes of NS patients and previous mouse models of NS. a–c** Scatter plots of correlation between gene expression changes in lesional skin of NS patients and lesional skin of *Spink5* cKO mice (**a**), Tg.*hKLK5* mice (**b**) and *Spink5*⁻/⁻ mice (**c**). Each dot corresponds to a gene pair. **d** Venn diagrams of differentially up-regulated genes (adjusted *P*-value < 0.05; log2 fold change > 1) and down-regulated genes (adjusted *P*-value < 0.05; log2 fold change < −1) in *Spink5* cKO mice skin (green) and NS patients' lesional skin (red). **e** Lollipop plot of the 20 most significantly enriched biological process GO terms within the differentially up-regulated and down-regulated genes shared between *Spink5* cKO skin and NS patients' lesional skin. The size of each circle is proportional to the GO *P*-value. The red or blue arrows next to each GO id number indicate enrichment in the up-regulated or down-regulated DEGs, respectively. **f** Heatmap of log2 fold change values (color bar) of selected differentially expressed genes within the GOs 'keratinization', 'cell population proliferation', 'proteolysis' and 'negative regulation of peptidase activity' enriched in the differentially expressed genes (DEGs) overlapping between *Spink5* cKO and NS patient skin transcriptomes. **g** Scatter plot of correlation between gene expression changes in lesional skin of *Spink5* cKO and Tg.*hKLK5* mice. **h, i** 2-D density plot of correlation between *Spink5* cKO and NS patients' lesional skin transcriptomes (**h**) and Tg.*hKLK5* and NS patients' skin transcriptomes (**i**) calculated only with the subset of genes annotated to the enriched GO terms related to protease activity ('proteolysis' and 'negative regulation of peptidase activity'). Several genes of interest are indicated with the corresponding human gene symbol. **j** Heatmap of log2 fold change values (color bar) of selected differentially expressed genes within the GO category "Immune and Inflammatory response" common to *Spink5* cKO and NS patients' lesional skin. Human gene symbols were used for annotation. **k** 2-D density plot of correlation between *Spink5* cKO and NS patients lesional skin transcriptomes calculated only with the subset of genes annotated to the GO terms within the "Immune and Inflammatory response" category. In (**a–c**), (**g–i**), and (**k**), Pearson correlation coefficient *r*, *P*-value and the total number of data points are indicated in the upper left corner of each plot. See also Supplementary Figs. 5–7 and Supplementary Data 1 and 2.

enriched in the shared up-regulated DEGs (Supplementary Fig. 6a, b). We found a high number of protease inhibitors to be differentially up-regulated in both *Spink5* cKO mice and NS patients' skin, such as *SLPI*, *SPINK7*, *CSTA*, *SERPINB3* and *SERPINB4* (Fig. 3f). Also, we detected high up-regulation of genes encoding several extracellular or transmembrane proteases such as *KLK13*, *MMP1*, *MMP10*, *TMPRSS11D*, *TMPRSS11A* and *PRSS27* (Fig. 3f and Supplementary Data 1).

Although the skin transcriptome profiles of Tg.*hKLK5* and *Spink5* cKO mice were very similar (Fig. 3g), the *Spink5* cKO skin transcriptome correlated significantly better to the lesional skin transcriptome of NS patients than did the Tg.*hKLK5* skin transcriptome (Fig. 3a, b, Supplementary Fig. 5g, and Supplementary Data 2). Since the skin barrier defect in Tg.*hKLK5* mice depends on the over-expression of human *KLK5*, despite intact levels of LEKTI, we hypothesized that the deregulation of genes encoding proteases or protease inhibitors in LEKTI-deficient skin will not be fully replicated in Tg.*hKLK5* skin. Accordingly, restricting the comparison to genes annotated in the "Protease activity" enriched GO group, the correlation of *Spink5* cKO and NS patient DEG pairs was significantly better than the correlation of Tg.*hKLK5* and NS patient DEGs (Fig. 3h, i and Supplementary Data 2). Therefore, although hyper-activation of KLK5 alone accounts for a great part of the skin genotype/phenotype in NS, LEKTI deficiency by itself (in the case of *Spink5* cKO mice) is a stronger inducer of NS skin disease. This is consistent with the fact that LEKTI inhibits the proteolytic activity of other epidermal proteases apart from KLK5.

Importantly, we identified that the most significantly enriched biological process GOs in the up-regulated DEGs shared between NS patient and *Spink5* cKO skin were GOs related to "Immune and Inflammatory response" (Fig. 3e and Supplementary Fig. 6a, c). Among the DEGs belonging to this GO group were genes involved in innate immunity, acute-phase response as well as genes coding for members of different cytokine families such as IL-20, IL-1, IL-36, or IL-17 and their downstream effector cytokines (Fig. 3j).

A comparison of *Spink5* cKO and NS patient skin gene expression profiles within the "Immune and Inflammatory response" GO group indicated a significantly better correlation than the comparison of their global gene expression profiles (Fig. 3k and Supplementary Data 2). This correlation was also significantly better than the correlations of gene expression profiles within the other groups of enriched GOs, except for the GO group "Cell migration" (Supplementary Fig. 6d–i and Supplementary Data 2), suggesting that inflammation response

and cell migration (probably related to inflammation response) are more likely to be conserved features of disease expression in *Spink5* cKO mice and NS patients.

Lastly, we extended the comparison of the *Spink5* cKO skin differential gene expression profile to that of other inflammatory skin diseases, namely psoriasis (PsO) and atopic dermatitis (AD) patients. The global differential gene expression profile of *Spink5* cKO skin resembles more that of psoriasis patients skin than that of AD patients skin (Supplementary Fig. 5h, i), which is consistent with previous comparisons of NS, psoriasis and AD skin transcriptomes[3]. Again, comparisons of differential gene expression within the "Immune and Inflammatory response" GO group showed significantly better correlation than the comparisons of global differential gene expression profiles, underscoring the major role of inflammation in shaping the common features of disease phenotype in *Spink5* cKO mice, NS, psoriasis and AD patients (Supplementary Fig. 6j, k and Supplementary Data 2).

In summary, lesional skin gene expression profiles of *Spink5* cKO mice and NS patients are most similar in terms of genes involved in inflammation response/signaling and protease activity. Furthermore, these skin transcriptome data reveal a conserved skin inflammation response between *Spink5* cKO mice and NS patients, which is characterized by IL-17, IL-36, and IL-20 family cytokine signaling pathways.

**Comparative skin proteome and transcriptome analyses of NS mouse models and NS patients confirm IL-36/IL-17 signaling and reveal novel endogenous substrates of KLKs.** Unrestrained epidermal serine protease activity in NS skin could contribute to post-translational regulation of gene expression through the proteolytic activation or degradation of secreted or cell surface proteins. This implies that skin transcriptome profiling may not suffice to fully describe the molecular landscape of lesional skin in NS. Therefore, in parallel to skin transcriptome analyses, we performed mass spectrometry-based bulk skin proteomics profiling of lesional skin in *Spink5* cKO mice (Supplementary Fig. 8a). In addition, the skin proteomes of Tg.*hKLK5* and *Spink5*⁻/⁻ mice were also analyzed (Supplementary Fig. 8b, c). The skin proteomes of all three NS mouse models were then compared to the previously described skin proteome of NS patient lesional skin[3] (Fig. 4a–c). Although not statistically significant, the skin proteome of *Spink5* cKO mice correlated best, among the three mouse models, with the skin proteome of NS patients (Supplementary Fig. 8d). GO enrichment analysis of the up-regulated and down-regulated differentially expressed proteins

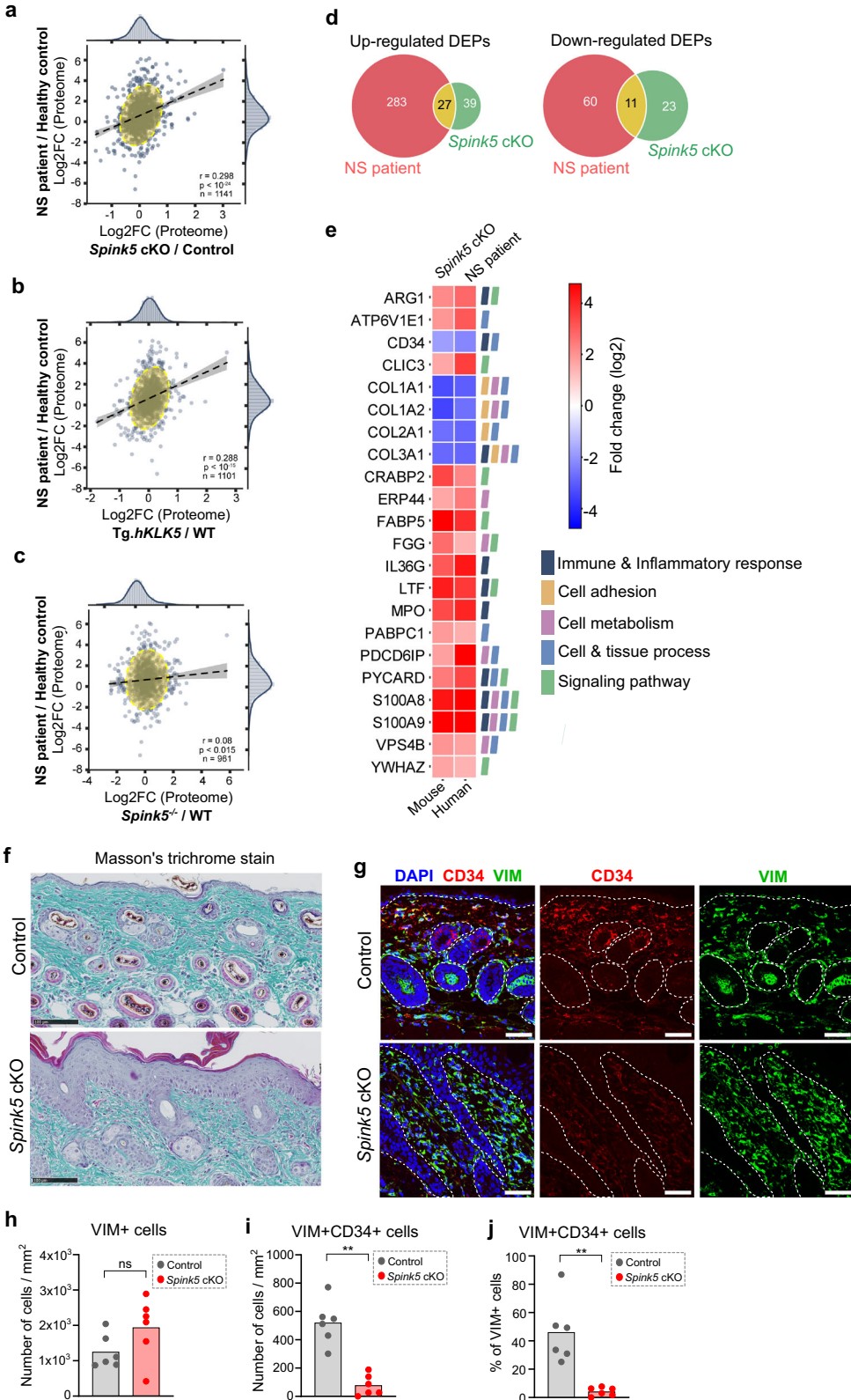

(DEPs) shared between *Spink5* cKO and NS patient skin proteome datasets (Fig. 4d) revealed that the most enriched biological process GOs grouped into "Cell adhesion", "Immune and Inflammatory response" and "Cell and Tissue process" (Supplementary Fig. 8e and Supplementary Data 3).

In the set of up-regulated DEPs shared between *Spink5* cKO and NS patient skin proteomes, S100A8 and S100A9 were among

the most highly expressed proteins (Fig. 4e, Supplementary Fig. 8f). Other shared up-regulated DEPs involved in skin inflammation were FABP5, MPO, LTF and IL-36G. These findings are in agreement with IL-36 and IL-17 signaling signatures identified in our skin transcriptomics analyses. The most down-regulated DEPs common to *Spink5* cKO and NS patient skin proteomes were several collagens and CD34 – a

**Fig. 4 Proteome profiling of *Spink5* cKO skin and comparison with skin proteome of NS patients and previous mouse models of NS. a–c** Scatter plots of correlation between proteins expressed in NS patient lesional skin and lesional skin of *Spink5* cKO mice (**a**), Tg.*hKLK5* mice (**b**), and *Spink5*⁻/⁻ mice (**c**). Each dot corresponds to a protein pair. **d** Venn diagrams of differentially up-regulated proteins (adj. *P*-value < 0.05 and log2 fold change > 1 for NS patients' skin samples; adj. *P*-value < 0.05 and log2 fold change > 0.33 for *Spink5* cKO skin samples) and down-regulated proteins (adj. *P*-value < 0.05 and log2 fold change <−1 for NS patients' skin samples; adj. *P*-value < 0.05 and log2 fold change <−0.33 for *Spink5* cKO skin samples) in *Spink5* cKO mice skin (green) and NS patients' lesional skin (red). **e** Heatmap of log2 fold change values (color bar) of differentially expressed proteins (DEPs) within the biological process GO terms that were significantly enriched in the set of DEPs shared between *Spink5* cKO mice and NS patient lesional skin. The color legend denotes the GO groups according to the similarity of the biological process of GO terms. **f** Masson's trichrome staining of back skin paraffin sections from control and *Spink5* cKO mice to visualize the density of collagen fibers (blue). Images are representative of analyses performed on samples from 6 different mice per group. Scale bar: 100 µm. **g** Confocal fluorescence microscopy images of CD34 (red) and Vimentin (VIM, green) double immunofluorescence staining in back skin paraffin sections of control and *Spink5* cKO mice. Nuclei are counterstained with DAPI (blue). The dermal-epidermal junction is outlined with a white dashed line. Scale bars: 50 µm. **h–j** Quantification of VIM+ and VIM+CD34+ cells in double immunofluorescence staining images of control and *Spink5* cKO skin. The number of VIM+ cells (**h**) and VIM+CD34+ cells (**i**) and the percentage of VIM+CD34+ cells from the total number of VIM+ cells (**j**) are shown. Each data point represents the average from the quantification of at least 3 images per skin sample from a different mouse with n = 6 mice per group. In **a–c**, Pearson correlation coefficient *r*, *P*-value and the total number of data points are indicated in the lower right corner of each plot. See also Supplementary Fig. 8 and Supplementary Data 3.

marker of several types of progenitor cells, including fibroblast progenitors and hair follicle bulge stem cells[40,41] (Fig. 4e). Using histological and immunofluorescence stainings, we confirmed the reduction of collagen fibers and CD34 expression in dermis of *Spink5* cKO mice (Fig. 4f–j and Supplementary Fig. 8g). Double staining of CD34 and the marker of fibroblasts Vimentin (VIM) revealed around 46% of CD34+VIM+ cells in dermis of control mice and only 4% of CD34+VIM+ cells in *Spink5* cKO dermis (Fig. 4h–j). Thus, the reduction of CD34+ fibroblasts could contribute to the reduction of collagen fibers in NS skin. Interestingly, loss of CD34+ fibroblasts has been observed in other inflammatory skin diseases[42]. The down-regulation of collagens and CD34 could also be related to increased matrix-metalloproteinase and KLK activities in lesional skin[43] and implies a possible hair growth defect (caused by depletion of CD34+ hair follicle progenitor cells) in addition to hair shaft abnormalities described in NS[44].

Hyper-activation of epidermal proteases is a hallmark of NS leading to enhanced degradation of extracellular proteins in the skin. Therefore, we hypothesized that such protein degradation will be reflected in the skin proteome of NS patients and NS mouse models and can be inferred by comparative analyses of the respective skin transcriptomes and proteomes. To test this hypothesis and to search for possible targets of hyperactive extracellular proteases, we performed correlation analyses of the fold change expression values for mRNA and protein pairs in skin of NS patients, *Spink5* cKO mice and Tg.*hKLK5* mice (Figs. 3 and 4). We first screened for mRNA-protein pairs that failed to correlate in both data sets (*Spink5* cKO and NS patient, or *Spink5* cKO and Tg.*hKLK5*) and then performed GO enrichment analyses on the resulting shared data sets (Fig. 5a, b and Supplementary Figs. 9a, b and 10). After selecting mRNA-protein pairs that belong to significantly enriched GO terms, we applied final criteria based on expression fold change and adjusted *P*-value (Supplementary Fig. 9c). We then focused on genes encoding secreted or membrane proteins, whose protein expression levels were lower than the expected, thus being potential substrates of active extracellular proteases in the skin (Fig. 5c, d and Supplementary Fig. 9d, e). In the transcriptome/proteome comparison of *Spink5* cKO and NS patient skin, we found a reduction from the expected protein level of the plasma membrane protein EPB41L3 and the extracellular glycoprotein AHSG (also known as Fetuin-A) (Fig. 5c). Importantly, when comparing the skin transcriptome and proteome of *Spink5* cKO and Tg.*hKLK5* mice, we observed again a significant reduction of AHSG. This last comparison also yielded several other potential protease substrates such as the extracellular proteins FETUB (also

known as Fetuin-B), IL-36A, NUCB2, and PZP and the plasma membrane protein CKAP4 (Fig.5d).

AHSG was previously identified as a putative substrate of KLK14[45]. Here, using in vitro recombinant protein digestion assay, we validated AHSG as a substrate of human KLK14 (Fig. 5e and Supplementary Figs. 12, 21, and 23). Among the epidermal KLKs tested, AHSG seems to be a KLK14-specific substrate, since recombinant human KLK5, KLK6, KLK7, KLK8 or KLK13 did not cleave AHSG (Supplementary Figs. 11a–c, 12, and 23). Interestingly, AHSG (Fetuin-A) and its paralogue FETUB (Fetuin-B) are endogenous inhibitors of meprin α and meprin β metalloproteinases[46,47], while meprin β is known to proteolytically activate KLK7[48]. Thus, decrease of AHSG and FETUB protein levels in the skin of NS mouse models and NS patients could indirectly contribute to KLK7 hyper-activation.

Next, we were interested to confirm whether human recombinant IL-36A can also be cleaved by epidermal proteases, since *Il36a* is among the genes whose protein expression level decreases in *Spink5* cKO and Tg.*hKLK5* skin, despite high mRNA expression fold changes (Fig. 5d). Furthermore, IL-36A protein could not be detected in NS patient skin by both mass spectrometry-based and immunofluorescence proteomic analyses, despite its high mRNA expression level[3]. Using in vitro recombinant protein digestion assay, we found that IL-36A is degraded by KLK14 (Fig. 5f and Supplementary Figs. 11d, 12, 21, and 23). KLK7, KLK5 and TMPRSS11D could also cleave human recombinant IL-36A, but much less efficiently (only after overnight incubation and at higher molar concentration of enzyme) (Supplementary Figs. 11e–g, l and 23). We also checked whether IL-36G – another member of the IL-36 pro-inflammatory cytokine family that is highly up-regulated at the mRNA and protein levels in skin of NS patients and mouse models – can also be proteolytically modified by epidermal proteases. Interestingly, KLK7 can cleave human recombinant full-length IL-36G resulting in an intense band of about 16 kDa, suggesting possible cytokine proteolytic activation (Supplementary Figs. 11h, 21, and 23)[49]. KLK14, KLK5 and TMPRSS11D can also cleave IL-36G, but much less efficiently (only after overnight incubation and at higher molar concentration of enzyme) (Supplementary Figs. 11h–l and 23).

Collectively, these proteomics analyses confirm IL-36 and IL-17 signaling signatures in both *Spink5* cKO and NS patient skin. Through comparative analyses of skin transcriptomes and proteomes, we identified potential endogenous substrates of epidermal proteases / KLKs that could contribute to NS pathophysiology, as demonstrated by the KLK14-mediated cleavage of AHSG and IL-36A that we validated. Furthermore, we found a possible proteolytic activation of IL-36G mainly through KLK7.

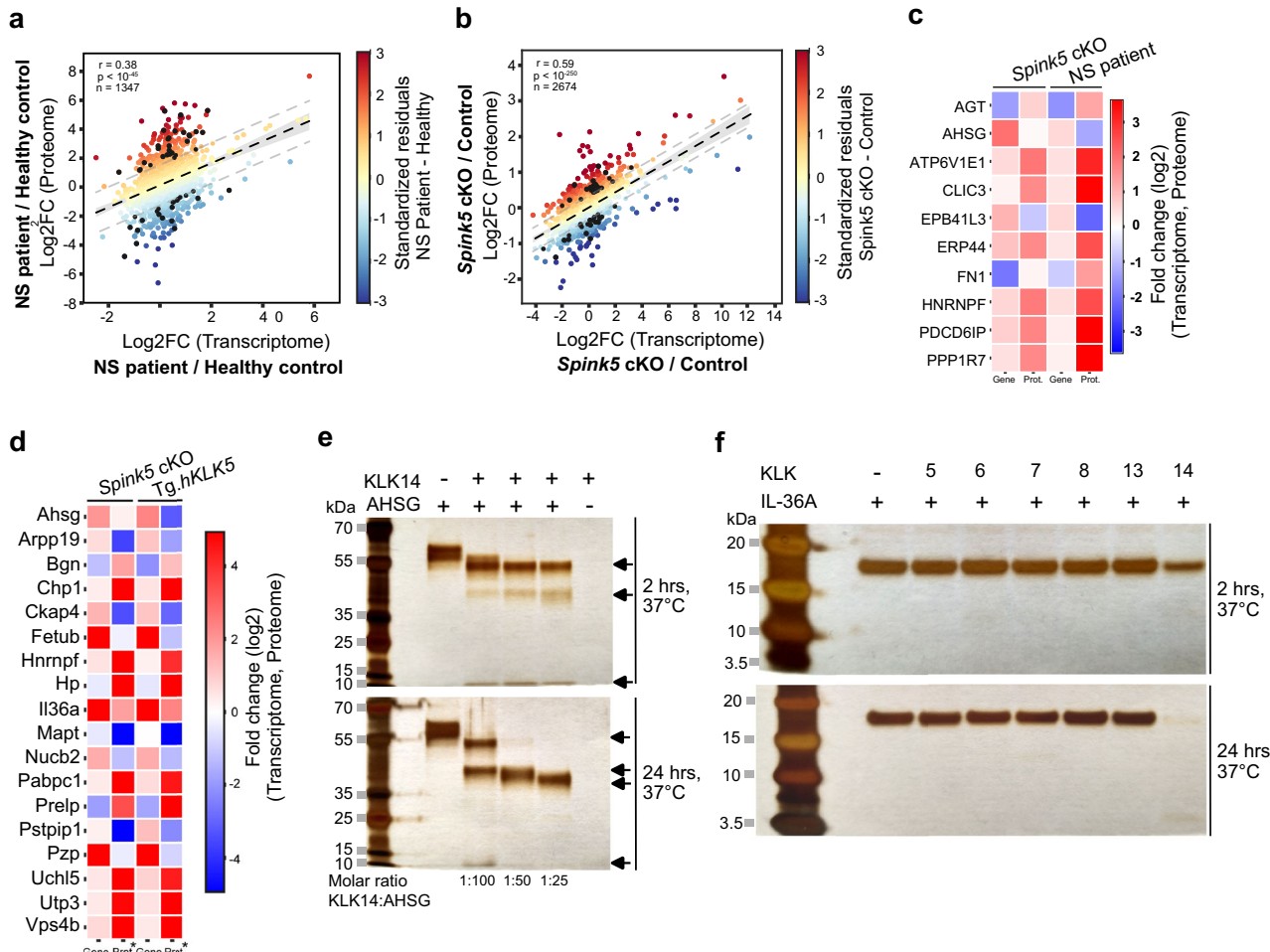

**Fig. 5 Comparative transcriptome and proteome analyses of *Spink5* cKO and NS patient lesional skin identify putative endogenous substrates of skin proteases. a**, **b** Scatter plot of correlation between mRNA and protein fold changes in NS patient (**a**) and *Spink5* cKO (**b**) lesional skin. Standardized residual values are superimposed on the scatter plots as a color gradient (color bar) and indicate the difference (standardized residual) between the observed protein fold change of each mRNA-protein pair and the expected protein fold change given by the correlation between mRNA-protein pairs. The regression line is represented as a black dashed line. Gray dashed lines mark the threshold values +1 and −1 of standardized residuals used as filter criteria for subsequent analyses. Black dots indicate the position of the selected mRNA-protein pairs common to NS patient and *Spink5* cKO transcriptome-proteome datasets, that belong to any significantly enriched GO term and whose standardized residual value is >1 or < −1 in both NS patient and *Spink5* cKO datasets. Pearson correlation coefficient *r*, *P*-value and the total number of data points are indicated on each plot. **c**, **d** Heatmap of log2 fold change expression values of mRNA-protein pairs, whose protein levels are significantly shifted from the corresponding transcript expression levels in both NS patient and *Spink5* cKO mouse skin (**c**), or in both *Spink5* cKO and Tg.*hKLK5* skin (**d**). **e** Silver stain SDS-PAGE gel analyses of protein fragments obtained by in vitro digestion of recombinant human full-length AHSG protein with recombinant human KLK14 enzyme at different enzyme to substrate molar ratios after incubation for 2 h and 24 h at 37 °C. Black arrows indicate the cleavage products. **f** Silver stain SDS-PAGE gel analyses of protein fragments obtained by in vitro digestion of recombinant human full-length IL-36A by recombinant human KLK5, KLK6, KLK7, KLK8, KLK13, or KLK14 at enzyme to substrate molar ratio of 1:100 after 2-h (upper panel) and overnight (lower panel) incubation at 37 °C. See also Supplementary Figs. 9–12.

**Innate and adaptive cellular immune responses shape the skin inflammation phenotype of *Spink5* cKO mice.** Analyses of NS patient and *Spink5* cKO skin transcriptomes and proteomes pointed out the importance of pro-inflammatory cytokines such as IL-36, IL-20 and IL-17 in NS skin inflammation. To confirm these omics data at the tissue/cellular level and to identify the different cell types that contribute to these inflammation signatures, we performed immunofluorescence staining and histological analyses of skin samples from *Spink5* cKO and WT control mice.

Several innate and adaptive immune cell types were increased in the skin of *Spink5* cKO mice such as mast cells (3-fold increase), neutrophils (9-fold), macrophages (1.6-fold), B cells (8.2-fold), T cells (2.5-fold), IL-17A+ cells (3.9-fold) and FOXP3+ cells (7.4-fold) (Fig. 6a–f and Supplementary

Fig. 13a–d). Co-staining of IL-17A and the pan-T cell marker CD3 revealed very few IL-17A+CD3+ cells, suggesting that immune cells other than Th17 cells could be producing IL-17A in *Spink5* cKO skin (Supplementary Fig. 13e–e′).

Next, we analyzed the expression level and pattern of pro-inflammatory cytokines in *Spink5* cKO skin. TSLP, a cytokine induced by PAR-2 receptor activation, had variable expression level within tissue and among individual mice and was not significantly increased in *Spink5* cKO skin (Supplementary Fig. 13f–f″). This finding is in contrast with the observations in *Spink5*[−/−] and Tg.*hKLK5* mouse models of NS reported previously[26,28]. Despite its possible degradation in vivo by KLK14, the pro-inflammatory cytokine IL-36A was up-regulated in the epidermis of *Spink5* cKO mice (4-fold increase), showing highest expression in the upper epidermal layers

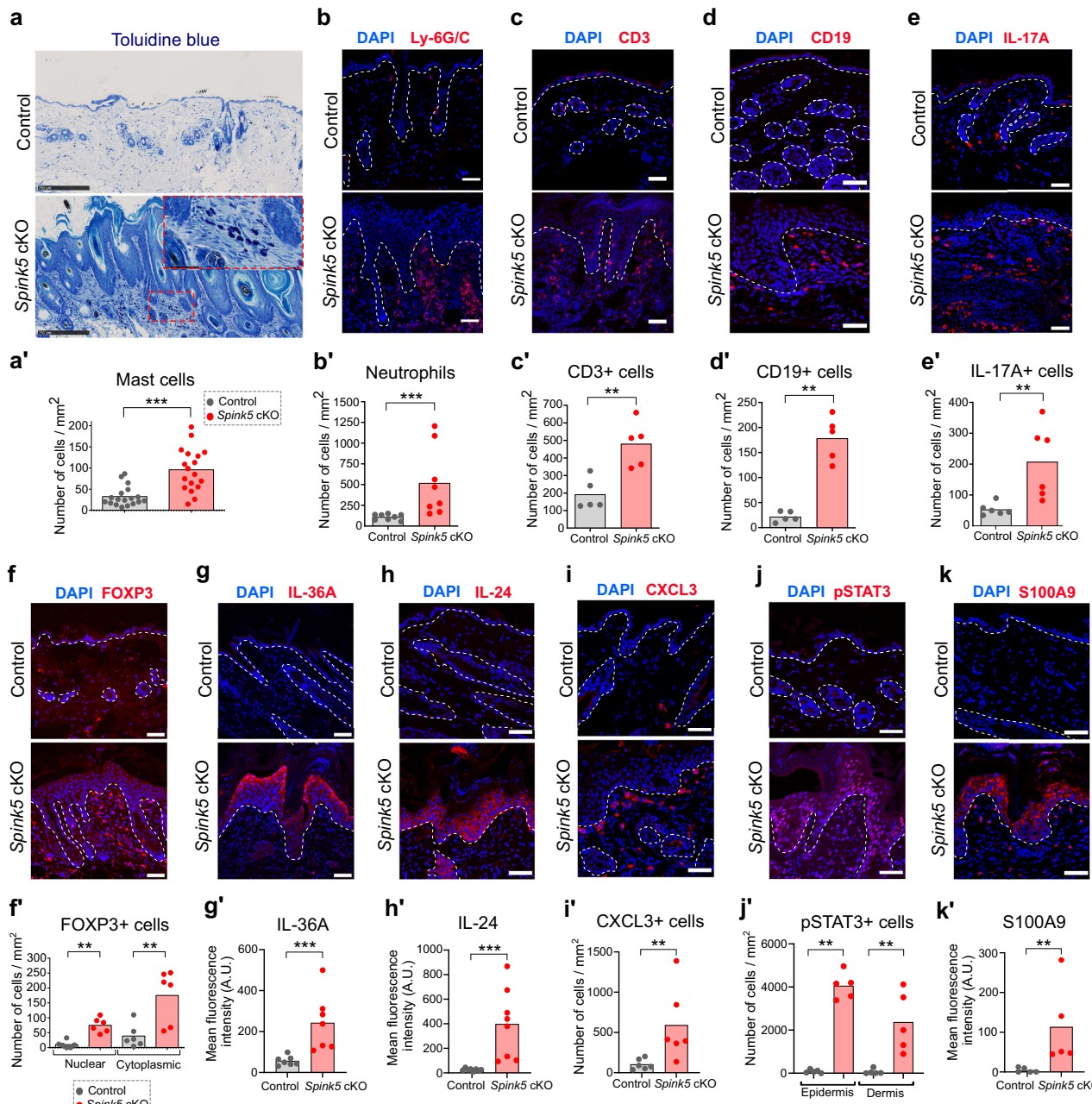

**Fig. 6 *Spink5* cKO mice display a multifaceted skin inflammation phenotype predominated by IL-36/IL-17 signaling. a** Detection of mast cells by toluidine blue staining (dark violet) of back skin paraffin sections from control (upper panel) and *Spink5* cKO (lower panel) mice. The zoomed area in the inset is outlined with a red dashed-line rectangle. Scale bars: 250 µm. **a'** Quantification of mast cell dermal infiltrates in control and *Spink5* cKO mice. **b–e** Immunofluorescence staining (red) of Ly-6G/C (**b**), CD3 (**c**), CD19 (**d**), and IL-17A (**e**) in back skin of control (upper panel) and *Spink5* cKO (lower panel) mice and quantification of neutrophils (Ly-6G/C+ cells, **b'**), T cells (CD3+ cells, **c'**), B cells (CD19+ cells, **d'**) and IL-17A+ cells (**e'**) in immunostained skin sections. Scale bars: 50 µm. **f–k** Immunofluorescence staining (red) of FOXP3 (**f**), IL-36A (**g**), IL-24 (**h**), CXCL3 (**i**), pSTAT3 (**j**), and S100A9 (**k**) in back skin sections of control (upper panels) and *Spink5* cKO (lower panels) mice. Scale bars: 50 µm. **f'** Number of cytoplasmic FOXP3+ cells (activated conventional T cells) and nuclear FOXP3 + cells (Treg cells) quantified in immunostained skin sections from control (gray) and *Spink5* cKO (red) mice. **g'–k'** Quantification of IL-36 (**g'**), IL-24 (**h'**), and S100A9 (**k'**) immunostaining signal in epidermis, number of CXCL3+ cells in dermis (**i**) and number of pSTAT3+ cells (**j'**) in dermis and epidermis of immunostained skin sections from control (gray) and *Spink5* cKO (red) mice. In (**b–e**) and (**f–k**), the dermal-epidermal junction is outlined with a white dashed line and nuclei are counterstained with DAPI (blue). Images are representative of immunofluorescence staining performed on skin samples of at least 5 different mice per group. Data in **a'–e'** and **f'–k'** are means (bars) and scatter plots, where dots correspond to values measured for individual mice ($n \geq 5$ per group). Statistical significance was determined using two-tailed unpaired non-parametric Mann–Whitney test: *$p < 0.05$ **$p < 0.01$, ***$p < 0.001$, ns (not significant). Control mice are *Spink5*$^{fl/fl}$ and/or *Spink5*$^{fl/-}$; *Spink5* cKO mice are *KRT14-CreERT2*$^{(Tg/0)}$/*Spink5*$^{fl/fl}$ and/or *KRT14-CreERT2*$^{(Tg/0)}$/*Spink5*$^{fl/-}$. See also Supplementary Fig. 13.

(Fig. 6g–g′). The expression pattern of IL-36A is consistent with its possible proteolytic regulation by epidermal KLKs, which are expressed in the same compartment. IL-24, another pro-inflammatory cytokine, was strongly increased (15-fold) in all epidermal layers of *Spink5* cKO mice (Fig. 6h–h′). CXCL3 – a major chemotactic factor for neutrophils and one of the top up-regulated DEGs identified in bulk RNAseq of *Spink5* cKO skin – showed strong expression in immune cells infiltrating the dermis and weak, variable expression in the epidermis (Fig. 6i–i′). The epidermis and dermis of *Spink5* cKO lesional skin displayed nuclear phosphorylated STAT3 expression (Fig. 6j–j′). Activation of JAK/STAT signaling downstream IL-36 and IL-24 has been previously reported in psoriasis[50,51]. Accordingly, we observed increased expression of the IL-36 downstream targets S100A8 and S100A9 in both epidermal keratinocytes and immune cell infiltrates in *Spink5* cKO skin (Fig. 6k–k′ and Supplementary Fig. 13i–k′).

Together these analyses indicate that *Spink5* cKO mice display a multiform skin inflammation featured by infiltration of neutrophils, IL-17A+ cells and B cells, keratinocyte-specific expression of IL-36A and IL-24 cytokines, and activated JAK/STAT3 signaling in epidermis and dermis.

**Innate immune response, IL-22/IL-17 signaling, splenic lymphocyte depletion and thymic atrophy are hallmarks of systemic inflammation in *Spink5* cKO mice.** Previous studies characterizing the blood of NS patients have not been sufficient to determine whether inherent immune system abnormalities contribute to NS pathology[4]. *Spink5* cKO mice being viable in adult stages allowed us to study the features of systemic inflammation response in NS by analyzing blood and lymphoid organs at the cellular and molecular levels. Similar to NS patients, *Spink5* cKO mice displayed increased serum IgE levels (Fig. 7a). Further analyses of blood revealed high levels (9-fold increase) of circulating neutrophils in *Spink5* cKO mice as compared to control littermates, which correlated significantly with skin phenotype severity (Fig. 7b and Supplementary Fig. 14c).

Lymph nodes in *Spink5* cKO mice were enlarged and had increased number of well-defined follicles and germinal centers as well as increased cellularity (Fig. 7c–c′ and Supplementary Fig. 14d). There were no significant differences in the frequency or absolute number of B lymphocytes, CD4+ or CD8+ T lymphocytes in lymph nodes of *Spink5* cKO and control mice (Supplementary Figs. 14a, b and 15a–d). However, we observed increased number of neutrophils (Ly-6G/C marker), S100A8+, S100A9+ and IL-17A+ cells in *Spink5* cKO lymph nodes (Supplementary Fig. 15e–h), which is consistent with the fact that neutrophils are known to express high levels of S100A8/9 proteins and can also produce IL-17A[52,53]. To gain further insight into the signaling pathways driving systemic inflammation in NS, we performed bulk RNAseq of inguinal lymph nodes from *Spink5* cKO and control mice (Fig. 7f, Supplementary Fig. 16a, b, and Supplementary Data 4). The up-regulation of innate immune response signature genes and IL-22/IL-17 signaling is in agreement with the increase of neutrophils and IL-17A+ cells in lymph nodes (Supplementary Fig. 15e–h). Interestingly, the gene coding for FGF23 – a hormone that regulates phosphate and vitamin D metabolism – was among the top up-regulated DEGs (fold change = 67) in *Spink5* cKO lymph nodes (Fig. 7f and Supplementary Data 4). The up-regulation of *Fgf23* in lymph nodes could be related to its function as a pro-inflammatory and immune-modulatory hormone[54]. The up-regulation of genes involved in neurotransmission could be related to the fact that lymph nodes are innervated by sensory neurons with immuno-modulatory role (Fig. 7f and Supplementary Data 4)[55]. Finally,

the down-regulation of genes involved in adaptive thermogenesis could be a compensatory mechanism for fever-induced energy expenditure (Fig. 7f and Supplementary Data 4)[56].

Since the skin of *Spink5* cKO mice displayed an inflammatory response gene expression signature associated with infiltration of immune cells, we searched for the common up-regulated and down-regulated DEGs in *Spink5* cKO skin and lymph node that could indicate the molecular pathways driving skin and systemic inflammation. Remarkably, the global skin and lymph node transcriptomes in *Spink5* cKO mice correlated poorly (Supplementary Fig. 16c) and the gene expression was much less altered in lymph nodes than in skin (209 DEGs in lymph nodes vs. 3471 DEGs in skin) (Supplementary Figs. 5a and 16a, d). The most significantly enriched biological process GO terms in the sets of overlapping up-regulated and down-regulated DEGs were related to inflammatory response, suggesting a migration of immune cells from the skin to the lymph nodes (Supplementary Fig. 16e and Supplementary Data 5). Some of the most up-regulated genes shared between skin and lymph node were involved in IL-22/IL-17 signaling, neutrophil degranulation, acute-phase response, or protease inhibition (Supplementary Fig. 16f–f′). Of note, type I IFN response genes were down-regulated in both *Spink5* cKO lymph nodes and skin (Fig. 7f and Supplementary Fig. 16e, f), which is in contrast with previous findings in NS patients[3]. Collectively, these data suggest that in *Spink5* cKO mice the main gene expression changes take place in the skin, while the gene expression changes observed in lymph nodes might reflect secondary events resulting from the systemic immune response to skin infection.

Next, focusing our analyses on the spleen, we observed that the spleen was enlarged in adult *Spink5* cKO mice ( > 6 weeks of age) mice, but had significantly reduced cellularity in young 4-week-old *Spink5* cKO mice (Fig. 7d–d′ and Supplementary Fig. 14e–g). Spleen weight did not correlate significantly with skin severity score neither in young nor in adult mice (Supplementary Fig. 14h, i). The enlarged spleens had a disorganized architecture with ill-defined white and red pulp, increased area of red pulp as well as absence of clear marginal zones surrounding follicles (Fig. 7d and Supplementary Fig. 14f). Flow cytometry analyses revealed that the number and frequency of splenic B cells was significantly decreased (by 2-fold) in young 4-week-old *Spink5* cKO mice compared to littermate controls (Supplementary Fig. 15i, j). The number of splenic T cells was also significantly decreased, but not their frequency (Supplementary Fig. 15k-l). Remarkably, we observed an increase of neutrophils (Ly-6G/C+ cells), S100A8+, S100A9+, and IL-17A+ cells in *Spink5* cKO spleens (Supplementary Fig. 15m–p), which is consistent with our previous findings in skin and lymph nodes. S100A8/9 and Ly-6G/C are known to be expressed in different subtypes of myeloid cells. The massive infiltrate of myeloid cells in *Spink5* cKO spleens and blood could explain spleen enlargement in *Spink5* cKO mice and the depletion of T and B lymphocytes. Moreover, the IL-17A+ cells in the spleen were not T cells as revealed by double staining of CD3 and IL-17A (Supplementary Fig. 15q). Thus, as in the skin, it is possible that also in the spleen of *Spink5* cKO mice the IL-17A producing cells are of myeloid origin, such as neutrophils.

Remarkably, we observed a drastic reduction of thymus size in *Spink5* cKO mice, which correlated with skin phenotype severity (Fig.7e–e′ and Supplementary Fig. 14j–l). The reduction of thymus cellularity and cortex area were due to depletion of CD4+CD8+ double-positive (DP) thymocytes (Fig. 7g, h and Supplementary Figs. 14a, b and 17a, b). The magnitude of DP thymocyte depletion in *Spink5* cKO mice correlated strongly with the severity of their skin phenotype (Supplementary Fig. 17c). Although *Spink5* cKO thymi showed arrest of thymocyte development at the DP stage, the few remaining DP thymocytes

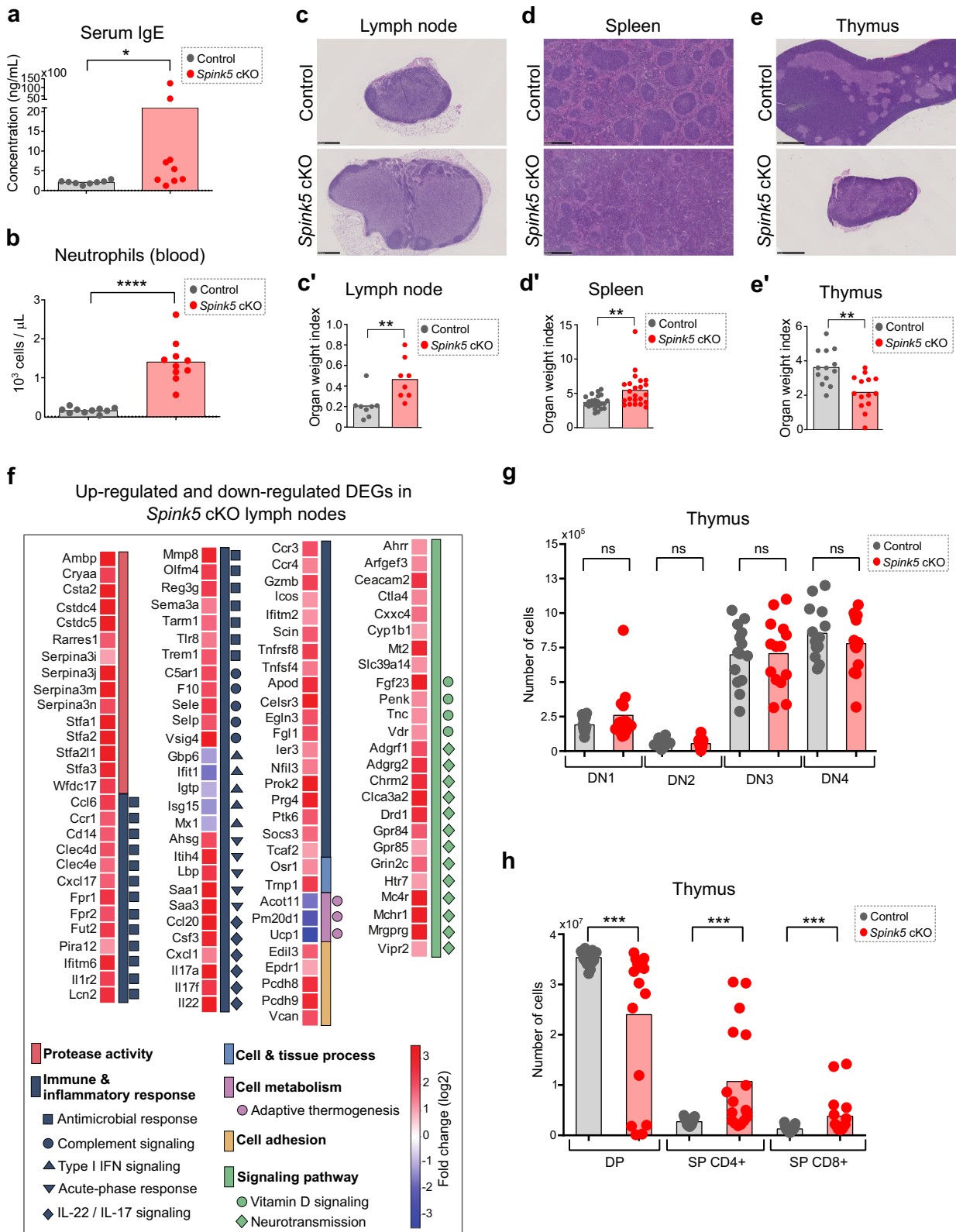

**a** Serum IgE

**b** Neutrophils (blood)

**c** Lymph node

**c'** Lymph node

**d** Spleen

**d'** Spleen

**e** Thymus

**e'** Thymus

**f** Up-regulated and down-regulated DEGs in *Spink5* cKO lymph nodes

**g** Thymus

**h** Thymus

could still undergo positive selection and accumulate as CD4+ or CD8+ single-positive (SP) thymocytes (Fig. 7h and Supplementary Fig. 17d).

LEKTI is known to be expressed in the thymic Hassall's corpuscles – structures formed by terminally differentiated medullary thymic epithelial cells (mTECs)[5,22]. In the *Spink5* cKO model, KRT14-CreERT2 can be expressed in mTECs, thus inducing *Spink5* deletion in the thymus (Fig. 1b). We therefore asked whether *Spink5* loss-of-function in the thymic medulla can affect thymocyte development and thus contribute to immuno-deficiency or autoimmune disease. To this aim, we performed several analyses: (1) quantification of FOXP3+T regulatory cells in thymic medulla, lymph node and spleen (Supplementary Fig. 17e, f), (2) measurement of *Aire* mRNA levels in the thymus

**Fig. 7 Systemic inflammation phenotype of *Spink5* cKO mice. a** Serum IgE levels measured by ELISA in control (gray) and *Spink5* cKO (red) mice. **b** Neutrophil counts measured in whole blood of control (gray) and *Spink5* cKO (red) mice. **c–e** Images of hematoxylin- and eosin-stained paraffin sections of inguinal lymph node (**c**), spleen (**d**), and thymus (**e**) from control (upper panels) and *Spink5* cKO (lower panels) littermate mice at an average age of 4.5 weeks. Scale bars: 500 μm (lymph node), 1 mm (spleen and thymus). **c′–e′** Quantification of the organ weight index (organ weight in milligrams divided by total body weight in grams) of control (gray) and *Spink5* cKO (red) mice. **f** Heatmap of log2 fold change values of differentially expressed genes in lymph nodes of *Spink5* cKO mice. This set of genes was annotated in the significantly enriched biological process GO terms (shown in Supplementary Fig. 16b) determined by analyzing the up-regulated (log2FC > 1 and adj. *P*-value < 0.05) and down-regulated (log2FC < −1 and adj. *P*-value < 0.05) genes in *Spink5* cKO lymph nodes. The color legend denotes the GO groups according to the similarity of the biological process of each GO term. Additional sub-groups of genes depending on their function are indicated by symbols. **g** Number of DN1 (CD25-CD44+), DN2 (CD25+CD44+), DN3 (CD25+CD44-) and DN4 (CD25-CD44-) thymocytes in the thymus of control (gray) and *Spink5* cKO (red) littermate mice determined by flow cytometry. **h** Number of DP (CD8+CD4+), SP CD4+ (single-positive CD4+CD8-) and SP CD8+ (single-positive CD4-CD8+) thymocytes in the thymus of control (gray) and *Spink5* cKO (red) littermate mice determined by flow cytometry. Data in (**a**, **b**), (**c′–e′**), and (**g**, **h**) are means (bars) and scatter plots, where data points correspond to the mean values measured for individual mice ($n \geq 8$ per group). Statistical significance was determined using two-tailed non-parametric Wilcoxon matched-pairs signed rank (**g**, **h**) or two-tailed unpaired non-parametric Mann-Whitney tests (**a**, **b**) and (**c′–e′**): *$p < 0.05$, **$p < 0.01$, ***$p < 0.001$, ****$p < 0.0001$, ns (not significant). Control mice are *Spink5*[fl/fl] and/or *Spink5*[fl/-]; *Spink5* cKO mice are *KRT14-CreERT2*[(Tg/0)]/*Spink5*[fl/fl] and/or *KRT14-CreERT2*[(Tg/0)]/*Spink5*[fl/-]. See also Supplementary Figs. 14–17.

(Supplementary Fig. 17g), (3) analysis of TCRα receptor repertoire in thymocytes (Supplementary Fig. 17h) and (4) measurement of T cell emigration factors in the thymus (Supplementary Fig. 17i, j). We did not detect any significant differences between control and *Spink5* cKO mice in the above analyses, suggesting that primary T cell immunodeficiency does not contribute to the chronic inflammation in *Spink5* cKO mice.

Acute infectious diseases and stress are known to cause thymic atrophy through depletion of immature CD4+CD8+ thymocytes[57,58]. A previous study reports intrathymic up-regulation of pro-inflammatory cytokines during the early stages of acute endotoxin-induced thymic atrophy[59]. We detected only a slight increase in the mRNA levels of *Tnf*, *Il6*, *Ifng*, *Cxcl1*, *Il17a*, and *Il22* in *Spink5* cKO thymi (Supplementary Fig. 17k–p). *S.aureus* bacteria frequently colonize the skin of NS patients, leading to disease exacerbation[2]. Thus, the above results, together with the high neutrophil counts in blood (Fig. 7b), suggest that the thymic atrophy in *Spink5* cKO mice might be a result of systemic bacterial infection. We confirmed by quantitative PCR of skin swab samples the presence of *S.aureus* in the skin of *Spink5* cKO mice (Supplementary Fig. 18a); however, culture of tissue extracts from skin and internal organs did not show significant differences in bacterial load between control and *Spink5* cKO mice (Supplementary Fig. 18b, c). Moreover, systemic treatment of *Spink5* cKO mice with broad-spectrum antibiotics cocktail, despite eliminating bacteria in internal organs and skin, did not reduce skin lesion severity, thymic atrophy and blood neutrophil counts (Supplementary Fig. 18d–k). Thus, the thymic atrophy in *Spink5* cKO mice is independent of skin or systemic bacterial infection and is most probably induced by stress and the resulting inflammatory milieu. The increased number of neutrophils in blood, spleen, lymph nodes and skin of *Spink5* cKO mice could be the result of increased serine protease activity that is known to induce sterile neutrophilic inflammation[60,61].

Since serine proteases such as KLKs are known to be expressed in the thymic medulla[62], where SPINK5 is also expressed[5,22], we were interested in analyzing the expression and activity of KLKs in the thymus of *Spink5* cKO mice. We observed a slight increase in the expression levels of several KLKs in *Spink5* cKO thymi (Fig. 8a–c and Supplementary Fig. 19a–h). Interestingly, the trypsin-like and chymotrypsin-like serine protease activities were significantly increased in thymus tissue extracts of *Spink5* cKO mice as compared to control littermates (Fig. 8d, e). Similar to thymic atrophy, the increase of serine protease activity in *Spink5* cKO thymi was not affected by systemic treatment with broad-spectrum antibiotics (Supplementary Fig. 19i, j) and the protease

activity levels correlated significantly with thymic atrophy (Supplementary Fig. 19k, l).

Collectively, the cellular and molecular analyses of blood and lymphoid organs in *Spink5* cKO mice reveal a systemic inflammation response marked by innate immune and IL-17/IL-22 signaling, granulocyte mobilization and depletion of splenic lymphocytes. Moreover, *Spink5* cKO mice displayed thymic atrophy characterized by depletion of CD4+CD8+ thymocytes and increased serine protease activity. Thymic atrophy and the increased blood neutrophil counts in *Spink5* cKO mice correlate significantly with skin lesion severity and are independent of bacterial infection.

## Discussion

In this study, we generated and analyzed a novel mouse model of NS by conditional ablation of *Spink5* in the epidermis based on a tamoxifen-inducible KRT14-CreER-mediated system. *Spink5* cKO mice develop NS-like skin phenotype and, being viable, allow to study NS disease for at least 3 weeks following induction of *Spink5* deletion.

*Spink5* cKO mice have the highest epidermal protease activity (relative to WT controls) as compared to previous viable mouse models of NS that over-express the human gene encoding *KLK5*[28] or *KLK14*[31]. This is expected since LEKTI deficiency renders uninhibited several epidermal proteases. Previous studies have demonstrated that genetic inactivation of both KLK5 and KLK7, but not either alone, is needed to fully reverse the skin barrier defect and inflammation phenotype in *Spink5* KO mice[63]. Our results suggest that KLK7 is the major active protease in *Spink5* cKO skin, showing higher relative activation (172-fold increase) as compared to the ensemble of trypsin-like serine proteases (19-fold increase). The contribution of KLK5 to the increase of trypsin-like serine protease activity in *Spink5* cKO skin is difficult to estimate with the current tools, since the GSK951A inhibitor used in our assay could inhibit, although less potently, other trypsin-like kallikrein-related proteases that might be present in the skin extract[64]. Interestingly, the trypsin-like serine protease and KLK7 activities measured in *Spink5* cKO skin relative to WT mice skin seem to be also higher than the relative serine protease activity measured in NS patient skin[15,36,65–67]. Our bulk skin RNAseq analyses identified several serine protease genes that are highly up-regulated in both *Spink5* cKO and NS patient lesional skin such as KLK13, PRSS27 (Marapsin), TMPRSS11A and TMPRSS11D (human air-way trypsin-like protease, HAT). Interestingly, previous studies have demonstrated that TMPRSS11D can proteolytically activate PAR-2 and contribute to

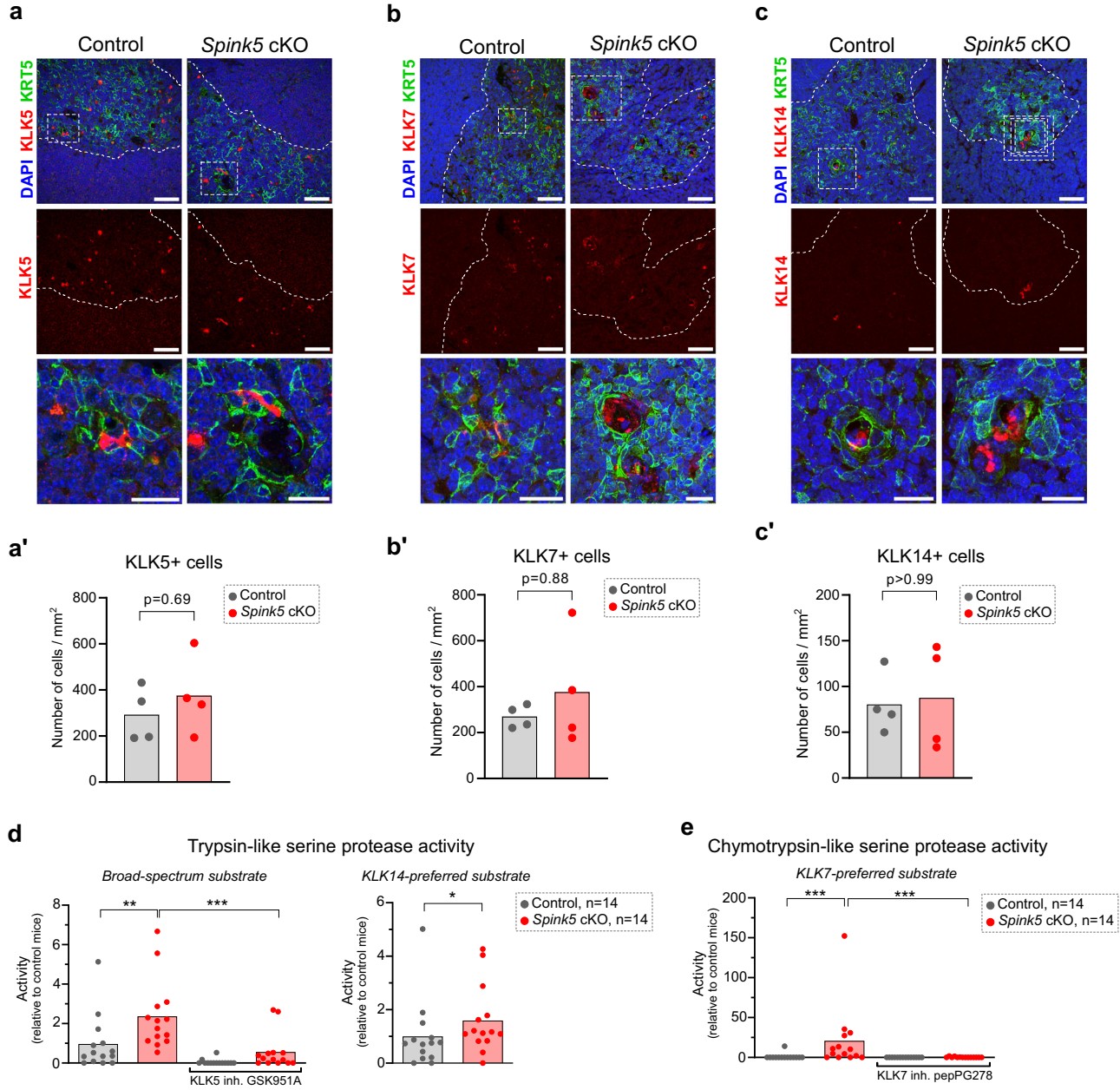

**Fig. 8 Expression and activity of kallikrein-related peptidases in *Spink5* cKO thymus. a–c** Double immunofluorescence staining of (**a**) KLK5 (red) and the marker of medullary thymic epithelial cells Keratin 5 (KRT5, green), (**b**) KLK7 and KRT5, and (**c**) KLK14 and KRT5 in paraffin sections of thymus from control and *Spink5* cKO mice. Thymic medulla is outlined by a white dashed line. The region marked with white dashed-line rectangles in the upper panels is the magnified image shown in the lowermost panels. Scale bars: 50 μm, magnified image: 25 μm. **a'–c'** Quantification of the number of KLK5+ cells (**a'**), KLK7+ cells (**b'**), and KLK14+ cells (**c'**) in immunofluorescence staining images of the thymic medulla region in control and *Spink5* cKO mice. Images are representative of immunofluorescence staining performed on thymus samples of at least 4 different mice per group. **d** Measurements of trypsin-like serine protease activity in protein extracts from thymus of control (gray) and *Spink5* cKO (red) mice using the broad-spectrum fluorogenic substrate for trypsin-like serine proteases Boc-VPR-amc (left panel) and the KLK14-preferred fluorogenic peptide substrate Ac-WAVR-amc (right panel). **e** Measurements of chymotrypsin-like serine protease activity in protein extracts from thymus of control mice (gray) and *Spink5* cKO mice (red) using the KLK7-preferred fluorogenic peptide substrate KHLY-amc. In **d** and **e**, activity is expressed as a ratio of the fluorescence intensity value measured in each thymus sample to the mean of the fluorescence intensity values measured in thymus extracts from control mice. The addition of the KLK5-specific inhibitor GSK951A or the KLK7-specific inhibitor pepPG278 serves as control to estimate the percent of trypsin-like or chymotrypsin-like protease activities due to KLK5 or KLK7 activation, respectively. Data in (**a'–c'**) and (**d**, **e**) are means (bars) and scatter plots, where dots correspond to values measured for individual mice (*n* ≥ 4 per group). Statistical significance was determined using two-tailed unpaired non-parametric Mann–Whitney test: *$p < 0.05$ **$p < 0.01$, ***$p < 0.001$, ns (not significant). Control mice are *Spink5*^fl/fl^ and/or *Spink5*^fl/-^; *Spink5* cKO mice are *KRT14-CreERT2*^(Tg/0)^/*Spink5*^fl/fl^ and/or *KRT14-CreERT2*^(Tg/0)^/*Spink5*^fl/-^. See also Supplementary Fig. 19.

inflammation in the lung epithelium and in skin of psoriasis patients[68,69]. Thus, apart from KLK5, KLK6 and KLK14 which are known to activate PAR-2[18], TMPRSS11D might also contribute to PAR-2 activation and subsequent skin inflammation in NS. So far, the roles of TMPRSS11D, TMPRSS11A, PRSS27 or KLK13 in NS pathophysiology have not been explored. Consistent with our findings, previous studies report that TNFα, IL-17A and the IL-20 subfamily cytokines can induce the expression of KLK6, KLK10, KLK11, KLK12, KLK13, KLK14 and PRSS27 and trypsin- and chymotrypsin-like activities in cultured normal human keratinocytes[70–72]. Our study reinforces the need to determine the redundancy, cross-regulation and activity profile of the epidermal proteases that contribute to skin barrier defect and inflammation in NS patients and mouse models.

The imbalance of protease activity that is induced by LEKTI deficiency could affect the proteolytic post-translational modification/regulation of key proteins in the skin. Here we applied an indirect, but unbiased, approach to uncover putative endogenous substrates of epidermal proteases. Interestingly, the targets that we retrieved with this comparative method were enriched in extracellular or membrane proteins and some of them had been previously identified as putative KLK substrates. For example, we identified NUCB2 (nucleobindin 2) as a potential target of proteolytic post-translational regulation. Interestingly, NUCB1, the paralogue of NUCB2, was previously identified as a putative substrate of KLK9[73] and KLK14[45]. Likewise, AHSG was previously identified as a putative KLK14 substrate in pancreatic cancer cell lines[45].

Here, we validated human AHSG as a substrate of human KLK14 using in vitro digestion assay. AHSG seems to be specific for KLK14, as the other skin KLKs tested in the same assay conditions did not cleave AHSG. AHSG is a secreted glycoprotein that belongs to the cystatin superfamily and has multiple functions[74]. Previous studies reported higher expression of AHSG in human fetal and newborn skin as compared to human adult skin[75]. Expression of AHSG in human adult skin is restricted to the proximity of hair follicles[75]. Ahsg-deficient mice display upregulation of pro-inflammatory genes such as Il12b and Il1b[76], which is consistent with the increased expression of Il12b and Il1b that we found in Spink5 cKO skin. Interestingly, homozygous missense mutations in human AHSG that disrupt phosphorylation or glycosylation sites needed for proper protein function are associated with alopecia and mental retardation syndrome[77]. Since KLK14 can cleave AHSG, KLK14 and AHSG are expressed in/near hair follicles[75,78], and transgenic mice over-expressing human KLK14 in the epidermis display prominent alopecia phenotype[31], KLK14-mediated degradation of AHSG might be involved in alopecia in NS. However, further analyses are needed to establish whether the fragments resulting from AHSG cleavage by KLK14 could retain any biological function.

Importantly, the pro-inflammatory cytokine IL-36A was also identified in our transcriptome/proteome comparative analyses as a candidate for post-translational regulation by proteolysis. Using in vitro digestion assay, we found that recombinant human KLK14 is efficiently degrading recombinant human full-length IL-36A. KLK5, KLK7 and TMPRSS11D could also cleave IL-36A, but much less efficiently. A previous study of transgenic mice over-expressing human KLK14 in the epidermis reports lower fold increase of IL-36A protein (3-fold) as compared to Il36a mRNA levels (23-fold) in skin of transgenic mice, further supporting a possible regulation of IL-36A protein by KLK14 in vivo[31]. The fact that KLKs and IL-36 cytokines are expressed in the same tissue compartment (epidermis) strongly suggests that KLKs could be involved in the regulation of IL-36 cytokine signaling in vivo. Thus, we extended our analyses and found that IL-36G, another member of the IL-36 cytokine family, was processed

very efficiently by KLK7 and to a lesser degree by TMPRSS11D and KLK14. Our results suggest a major role of KLK7 and KLK14 in regulating pro-inflammatory cytokine signaling in NS skin. Further biochemical analyses are needed to determine the nature and the biological activity of the proteolytic fragments resulting from IL-36A or IL-36G digestion by KLKs or TMPRSS11D. Likewise, future studies employing genetic knock-outs and degradomics methods are needed to confirm and extend our findings.

Our analyses of skin and lymph nodes in Spink5 cKO mice suggest IL-17/IL-22 signaling as the main mediator of systemic inflammation. This is consistent with a previous study of NS patient blood reporting the highest frequencies for Th17/Tc17 and Th22/Tc22 cells[79]. Although less pronounced, we also found some shared features of Th2/allergic inflammation signature in Spink5 cKO and NS patients, revealed by increased number of mast cells and expression level of CCL17 in skin and increased serum IgE levels. The secondary role of type 2 inflammation in our Spink5 cKO mice is consistent with the lack of efficacy of anti-IL-13 systemic treatment in another inducible Spink5 KO mouse model reported recently[80]. Surprisingly, we found unchanged levels of the allergy-associated cytokine TSLP in skin of Spink5 cKO mice, which contradicts previous studies with Spink5-/- mice, showing TSLP up-regulation in keratinocytes through KLK5-mediated PAR-2 activation[26]. This discrepancy could be due to the different state of inflammation in both models: early inflammation not modulated by exposure to pathogen or allergen in skin of newborn Spink5-/- mice and full-blown, chronic inflammation in skin of adult Spink5 cKO mice likely shaped by exposure to environmental factors such as microbiota and allergens. However, we found that systemic treatment of Spink5 cKO mice with broad-spectrum antibiotics cocktail does not rescue the neutrophil phenotype in circulation and the skin barrier defect. This sterile inflammatory response in Spink5 cKO mice could be driven by increased serine protease activity[60,61,81]. The proteolytic degradation/processing of extracellular structural or signaling proteins by KLKs, such as the IL-36 cytokines identified in this study, could produce chemoattractants for neutrophils. Interestingly, sterile neutrophilic inflammation is seen in other severe inflammatory skin diseases such as generalized pustular psoriasis, in which IL-36 signaling is a central driver of pathogenesis[82].

A hallmark of systemic inflammation in Spink5 cKO mice is thymic atrophy. Our findings do not support a role of systemic bacterial infection as a cause for thymic atrophy in Spink5 cKO mice. As the skin, the thymus of Spink5 cKO mice also displayed increased trypsin-like and chymotrypsin-like serine protease activities, which were sensitive to inhibition by KLK5- or KLK7-specific inhibitors. However, the signals measured with our assay were much lower as compared to the activity measured in skin extracts; therefore, the availability of more sensitive and KLK-specific protease substrates/probes should enable better characterization of protease activity in the thymus. While it is known that serine protease activity causes sterile neutrophilic inflammation, we don't know whether serine protease activity can cause thymic atrophy, through, for example, proteolytic activation of pro-inflammatory factors. Interestingly, only two previous studies report thymic atrophy, identified by post-mortem analyses of organs, in patients with severe congenital erythroderma[83,84]. Some of these patients were later diagnosed with Netherton syndrome. We exclude the presence of primary T cell immunodeficiency in Spink5 cKO mice, because the expression level of Aire was unchanged, there were no abnormalities in the number of T regulatory cells in the thymus, spleen or lymph nodes, and there were no changes in the TCRα repertoire. However, severe thymic atrophy can diminish the pool of T lymphocytes and, in

the long term, contribute to T-cell imbalances that could lead to acquired immunodeficiency. Our results call for more detailed analyses of thymus in both NS patients and mouse models with thymus-specific *Spink5* deletion.

Previous studies in NS patients identify skin and systemic inflammation signatures that justify the repurposed use of biotherapies targeting IL-17, IL-4, IL-23, or TNFα signaling pathways; however, the efficacy of these biologic therapies is variable[85]. Inhibition of the KLK/IL-36 signaling axis uncovered here could represent a promising therapeutic strategy that targets early pro-inflammatory events in NS skin. The viable *Spink5* cKO mouse model in combination with other genetic knock-out mice or pharmacological approaches represents a valuable tool to assess the relevance of these signaling pathways for NS pathology.

In conclusion, our study validates *Spink5* cKO mice as the model that most closely resembles NS, defines the molecular features of skin barrier defect and inflammation in mouse models of NS and in NS patients and uncovers potential novel signaling pathways interactions that could guide future research on therapeutic targets and disease mechanisms.

## Methods

**Mouse models**. Mice were housed in a specific pathogen-free animal facility at Imagine Institute (Paris, France) in constant environmental conditions and in individually ventilated cages with food and drinking water ad libitum. Mouse colony maintenance and experimental procedures were performed in accordance with the ethical standards set by the national legislation and revised by the ethical committee for animal experimentation at Imagine Institute. We have complied with all relevant ethical regulations for animal use. Tg.hKLK5 mice have been described previously[28]. Tg.hKLK5 mice were maintained as heterozygotes by breeding transgenic mice to wild-type mice of pure C57BL/6JRj genetic background. Skin samples were collected from 3-week-old transgenic mice and non-transgenic (wild-type) littermates of both sexes. *Spink5*$^{-/-}$ mice have been described previously[14]. *Spink5*$^{-/-}$ mice were bred in the C57BL/6JRj genetic background. Homozygous *Spink5* mutant mice were obtained by intercrossing *Spink5*$^{+/-}$ mice. For skin sample collection, newborns were euthanized as soon as possible after birth, since *Spink5*$^{-/-}$ mice survive for a few hours only.

*Spink5* conditional knock-out mice (*Spink5* cKO) were generated with technical services provided at Institut Clinique de la Souris – ICS, Strasbourg, France. The gene targeting strategy consisted in inserting loxP sites that flank *Spink5* exon 3 by homologous recombination in mouse embryonic stem cells (Supplementary Fig. 1). Mice with germline transmission of *Spink5* floxed alleles (*Spink5*$^{fl/fl}$) were backcrossed to the C57BL/6JRj genetic background. *Spink5*$^{fl/fl}$ mice were bred to KRT14-CreERT2$^{(Tg/0)}$ transgenic mice[33] to obtain KRT14-CreERT2$^{(Tg/0)}$/ *Spink5*$^{fl/fl}$ (*Spink5* cKO) mice. *Spink5* cKO mice were maintained in the C57BL/6JRj genetic background. Cre recombinase activity was induced by intraperitoneal injection of tamoxifen (Sigma, #T5648) (1 mg in 100 μL corn oil) once daily for 5 consecutive days or by topical application of 4-hydroxytamoxifen (Sigma, #H6278) (0.2 mg in 100 μL ethanol) on shaved back skin once daily for up to 5 consecutive days. We did not see differences in the outcome between topical and intraperitoneal tamoxifen-mediated induction of CreERT2. Topical induction of CreERT2 was preferred since it is non-invasive. We found that CreERT2-mediated *Spink5* floxed exon excision and subsequent development of NS-like skin phenotype is more efficient in mice that already show some basal level of spontaneous CreERT2 activity (assessed by the presence of the excised allele). Thus, we routinely screened all mice for spontaneous, tamoxifen-independent floxed allele excision. In mice with mild signs of phenotype due to tamoxifen-independent CreERT2 activation (e.g. only visible as slight scaling on the ears or on the muzzle), shorter induction (with 2–3 applications of topical 4-hydroxytamoxifen) was sufficient to induce whole-body knock-out phenotype even in mice older than 4 weeks. In such mice, topical 4-hydroxytamoxifen applied on the back skin could also induce CreERT2 activity in the skin of the belly, face, or extremities and in internal organs where the KRT14 promoter is active, such as esophagus, tongue, or thymus.

Female and male *Spink5* cKO mice and littermate controls at the age of 4–12 weeks were used in this study.

For mouse genotyping, genomic DNA was isolated from tail biopsies using NucleoSpin Tissue Mini kit (Macherey-Nagel). PCR for screening of *Spink5* conditional and knock-out alleles, *Spink5* constitutive knock-out allele and KRT14-CreERT2 transgene was carried out using allele-specific primers, Go Taq polymerase (Promega) and a Bio-Rad thermal cycler. We provide the sequences of the primers used for genotyping of *Spink5* cKO mice and the size of the resulting PCR products corresponding to the detected alleles (Table 1). The internal control primers specific for Cpxm1 were used together with the KRT14-CreERT2 specific primers in the PCR assay for detection of the KRT14-CreERT2 transgene (Table 1).

**Scoring of skin lesion severity**. The scoring system used to assess the extent and severity of skin lesions in *Spink5* cKO mice was based on the EASI score which gives 50% weight to intensity and 50% weight to extent of lesions[86]. The body was divided into four regions (head/neck/ears, back, abdomen, and extremities/tail) and the extent of skin lesions covering each of these body parts was scored from 0 to 6, a score of 0 being absence of lesions and a score of 6 being 100% of body area covered with lesions. For each body region, the severity of 5 signs of skin lesion (redness, edema/

**Table 1 Sequences of primers used for genotyping of *Spink5* cKO mice and the size of the corresponding PCR products.**

| Allele | Primer sequence (5′-3′) | PCR product size (bp) |
|---|---|---|
| *Spink5* conditional (floxed) allele | Fw – GAGGTCACTGGTAGCATCCATGAAA | 259 (floxed allele) |
| | Rv – AGCTATTTGGTCAGTTCAATAACAC | 193 (WT allele) |
| *Spink5* knock-out allele (after exon 3 excision) | Fw – GAGGTCACTGGTAGCATCCATGAAA | 1240 (floxed allele) |
| | Rv – TCCTTTGAAATCCCATTGAGTGATC | 192 (knock-out allele, Δex3) |
| *Spink5* constitutive knock-out allele | 5UTRSpink5 – GTTCTCAAGGAGTCTAACAT | 400 (knock-out allele) |
| | Ex1Spink5 – CTTGTGTGAGATAAAATGCC | 200 (WT allele) |
| | NeoR – ATGCGAAGTGGACCTGGGAC | |
| KRT14-CreERT2 | Fw – CCATCTGCCACCAGCCAG | 281 |
| | Rv – TCGCCATCTTCCAGCAGG | |
| Cpxm1 | Fw – ACTGGGATCTTCGAACTCTTTGGAC | 420 |
| (Internal control) | Rv – GATGTTGGGGCACTGCTCATTCACC | |

lichenification, excoriation, scaling, and alopecia) was evaluated on a scale of 0 (absence of sign) to 3 (severe).

**Measurement of transepidermal water loss**. Transepidermal water loss (TEWL) was measured on the lower back skin region of each mouse using the Tewameter TM300 device (Courage + Khazaka, Cologne, Germany). When necessary, the back skin region was shaved before each measurement. For each measurement, three consecutive readings of TEWL were made on the same back skin region and the average value was calculated.

**Systemic treatment of mice with antibiotics cocktail**. *Spink5* cKO mice and WT control littermates at the age of 4–5 weeks that had tamoxifen-independent CreERT2 activation, but no or mild/localized skin phenotype were used for this experiment (see scheme of treatment protocol in Supplementary Fig. 18d). A total of 6 mice were included in each study group. *Spink5* cKO and WT control littermate mice were treated with broad-spectrum antibiotics cocktail (ampicillin 0.5 g/L, streptomycin 1 g/L and metronidazole 0.5 g/L) in 2% sucrose-containing drinking water for a duration of 2 weeks. The placebo groups received 2% sucrose in drinking water. Treatment was started at the same time as the induction of CreERT2 by topical application of 4-hydroxytamoxiefen on shaved back skin (0.2 mg/mouse per day for a duration of 3 days). The antibiotics- or sucrose-containing water was changed with a freshly prepared solution once a week. Phenotype assessment (skin score, body weight, and transepidermal water loss measurements) were performed before treatment, at day 7 and at day 14 (end) of the treatment protocol. At the end of the treatment, euthanized mice were dissected under sterile conditions in laminar flow cabinet. Samples from skin and internal organs (spleen, lung and liver) were collected, weighed and placed in sterile 2 mL lysing matrix D tubes (MP Biomedicals) containing 0.5 mL of sterile PBS and a ¼" ceramic sphere, and subsequently homogenized using FastPrep-24 homogenizer (MP Biomedicals). Blood was collected by cardiac puncture and used for hematology analyses to quantify neutrophils. Thymus was harvested, weighed and processed for histology and/or protease activity measurement.

**RNA sequencing and data analyses**. RNA sequencing of back skin samples was carried out on five *Spink5* cKO (*KRT14-CreERT2*[(Tg/0)]/*Spink5*[fl/fl] and/or *KRT14-CreERT2*[(Tg/0)]/*Spink5*[fl/-]) and five control (*Spink5*[fl/fl] and/or *Spink5*[fl/-]) mice with a mean age of 5 weeks, five Tg.*hKLK5* and five WT (non-transgenic) littermate control mice aged 3 weeks, and four *Spink5*[-/-] newborn mice and four *Spink5*[+/+] control littermate newborn mice. RNA sequencing of lymph node tissue samples was carried out on five *Spink5* cKO (*KRT14-CreERT2*[(Tg/0)]/*Spink5*[fl/fl] and/or *KRT14-CreERT2*[(Tg/0)]/*Spink5*[fl/-]) and five control (*Spink5*[fl/fl] and/or *Spink5*[fl/-]) mice with a mean age of 5 weeks.

RNA was extracted from flash-frozen tissue samples using RNeasy Firbous Tissue kit (Qiagen). RNA quality and integrity were assessed using TapeStation 4150 instrument and RNA 6000 Nano Kit (Agilent Technologies). Sequencing libraries were prepared using total RNA with RIN value > 8.0 and a TruSeq stranded mRNA kit (Illumina). Whole transcriptome sequencing of tissue samples from *Spink5* cKO and Tg.*hKLK5* mice was performed using NovaSeq 6000 instrument (Illumina). The sequencing run was paired-end with read length $2 \times 150$ bp. For the skin samples of *Spink5*[-/-] mice and the corresponding wild-type controls, the whole transcriptome sequencing was performed using HiSeq2000 instrument (Illumina). The sequencing run was single-end with read length $1 \times 50$ bp.

The RNAseq data for the skin and lymph nodes samples of *Spink5* cKO mice and skin samples of Tg.*hKLK5* mice was analyzed using RASflow RNAseq analysis workflow[87]. Trim Galore was used to remove adapter sequences and low-quality sequences. Sequence reads were mapped to the mouse reference genome mm10 using HISAT2 program. The RNAseq data for the skin samples of *Spink5*[-/-] mice was processed using Illumina pipeline software. Reads were cleaned of adapter sequences and low-quality sequences using an in-house program (https://github.com/baj12/clean_ngs). The TopHat program (version 1.4.1.1, default parameters) was used to align the reads to the reference mouse genome mm10. Data normalization and differential expression analysis for all mouse model RNAseq datasets were performed using DESeq2[88]. The transcriptomic data were deposited at the Gene Expression Omnibus (GEO) data repository under accession number GSE224280 (*Spink5* cKO skin transcriptome), GSE224409 (*Spink5* cKO lymph node transcriptome), GSE224673 (Tg.*hKLK5* skin transcriptome), and GSE224674 (*Spink5*[-/-] skin transcriptome).

The RNAseq datasets for lesional skin of NS patients[3], psoriasis patients[89] and atopic dermatitis patients[89,90] have been described previously. The differential expression analyses of patient RNAseq datasets used in this study were performed using DEseq2. Differentially expressed genes were defined by a log2 fold change value higher than 1 or lower than −1 and adjusted *P*-value lower than 0.05.

For the comparisons of mouse datasets with human datasets, pairs of mouse-human orthologues were searched using the HomoloGene database (https://www.ncbi.nlm.nih.gov/homologene) and a customized Python script based on Pandas. Correlation analyses and Venn diagram analyses were performed in Python using stats from scipy, seaborn and matplotlib.pyplot libraries, and matplotlibvenn library, respectively. Data analysis and management were done by applying python scripts based on pandas[91], SciPy[92] and online tools using GO BioLink API (http://api.geneontology.org/api). Custom codes are fully available upon request. Data plots and heatmaps were created in Python using Pandas, Seaborn and matplotlib.pyplot and subsequently formatted using Adobe Illustrator.

**Mass spectrometry data acquisition and processing**. Mass spectrometry-based proteomics profiling of skin was performed using back skin samples collected from the same mice that were included in the RNAseq analyses. Frozen skin was cut and protein sample preparation was performed as described previously[3]. The *Spink5*[-/-] mouse samples were prepared label-free and the LC-MS/MS measurements were performed using data-dependent acquisition. For the *Spink5* cKO and Tg.*hKLK5* samples, the peptide concentration was measured after peptide clean-up using BCA assay and 25 μg of each sample was vacuum-concentrated until dryness. Those samples were TMT16-plex labeled, fractionated, and concatenated as previously described[93,94]. Concatenated fractions were re-suspended in 40 μl buffer A containing 0.1% formic acid in water (Honeywell) and sonicated for 5 min. For LC-MS/MS measurements, 300 ng of each fraction were separated using a 90 min separation gradient as previously described[94]. Raw data were analyzed using MaxQuant (v1.6.14.0) and a mouse proteome database (reviewed sequences, downloaded from uniprot.org on July 2nd, 2020, containing 17052 entries and 11 additional sequences for the iRT peptides). Further analysis and post-processing were performed as previously described[94].

Differential expression analysis was performed using limma (v3.40.0). Criteria for differentially expressed proteins in the *Spink5* cKO and Tg.*hKLK5* datasets were log2 fold change values higher than 0.33 or lower than −0.33 and adjusted *P*-value < 0.05.

In the *Spink5*$^{-/-}$ and NS patient lesional skin datasets, the criteria for differentially expressed proteins were log2 fold change values higher than 1 or lower than −1 and adjusted *P*-value < 0.05. Raw spectral files and intermediary identification search results were uploaded to the MassIVE repository (part of the ProteomeX-change consortium). Data can be accessed using the MassIVE accession number MSV000091184 or the ProteomeXchange accession number PXD039796.

Generation of proteome data for NS patient lesional skin by data-independent acquisition LC-MS/MS has been described previously[3].

**Gene ontology enrichment analyses**. Biological process gene ontology enrichment analyses were performed separately on the set of differentially up-regulated or down-regulated genes or proteins common to *Spink5* cKO skin and NS patient lesional skin using DAVID database[95,96]. Biological process gene ontology (GO) terms with Bonferroni adjusted *P*-value < 0.1 were then summarized using REVIGO[97]. The resulting GO terms with adjusted *P*-value < 0.05 were considered significantly enriched. Cellular component GO enrichment analyses (for the comparisons of skin transcriptome and proteome) were performed using DAVID database without REVIGO summary step and GO terms with Bonferroni adjusted *P*-value < 0.05 were considered significantly enriched.

**Skin transcriptome/proteome correlation analyses**. NS patient and mouse model lesional skin proteomes and/or transcriptomes were compared by correlation analysis using Python. Fisher's r-to-z transformation was used to determine whether any Pearson correlation was significantly better. To identify mRNA-protein pairs shared between NS patient skin and *Spink5* cKO skin or between *Spink5* cKO skin and Tg.*hKLK5* skin, in which the protein expression level is different from the predicted protein level (as indicated by the transcriptome-proteome correlation), we performed analyses consisting of 4 filtering steps. First, from the proteome-transcriptome correlation of each dataset (NS patient skin or mouse model skin), we selected proteins, whose expression level is shifted from the predicted protein expression given their transcript expression level by applying criteria based on standardized residuals with values higher than 1 and lower than −1 standard deviation. The resulting datasets were used for the second filtering step, in which we compared whether any selected element (mRNA-protein pair) in the *Spink5* cKO dataset was also selected in NS patient transcriptome/proteome or in Tg.*hKLK5* transcriptome/proteome datasets. In the third filtering step, the overlapping mRNA-protein pairs from each comparison (*Spink5* cKO vs NS patient and *Spink5* cKO vs Tg.*hKLK5*) were subjected to GO enrichment analysis using DAVID tool. The GO enrichment analyses were performed separately for down-regulated proteins (lower than −1 standard deviation) and up-regulated proteins (higher than 1 standard deviation). Both biological process and cellular component GO enrichment analyses were performed. The sets of proteins belonging to significantly enriched GO terms in either GO category were selected. Lastly, a fourth filtering step was applied, in which a significant shift of protein expression in the mRNA-protein pairs was determined based on fold change expression ratios and adjusted *P*-value (Supplementary Fig. 9c).

**RT-qPCR**. Snap frozen tissue samples (back skin and thymus tissue fragments of approximately 10 mg) were homogenized in RLT lysis buffer using Qiagen Tissue Lyser LT instrument. Total RNA was subsequently extracted using RNAeasy Fibrous Tissue Kit (Qiagen). cDNA synthesis was done using RevertAid H

**Table 2 Sequences of gene-specific primers (SYBR green assay) and references of pre-designed TaqMan gene expression assays used for qPCR.**

| Gene | Primer sequence (5′-3′) / TaqMan probe id |
|---|---|
| *Spink5* | Fw – GTCCCAAAGGCAACAATTCTTC |
| | Rv – ATGGGAAAACATACCGCAGTAG |
| *Tnf* | Fw – CCCCAAAGGGATGAGAAGTT |
| | Rv – CACTTGGTGGTTTGCTACGA |
| *Il6* | Fw – TAGTCCTTCCTACCCCAATTTCC |
| | Rv – TTGGTCCTTAGCCACTCCTTC |
| *Cxcl1* | Fw – GCCTATCGCCAATGAGCTG |
| | Rv – AAGGGAGCTTCAGGGTCAAG |
| *Klk6* | Fw – TGGGGAAACACAACCTACGG |
| | Rv – GGGGATGGACAATAGTCCTGT |
| *Klk8* | Fw – GATCCTGGAAGGTCGAGAGTG |
| | Rv – CTGCTCCGGCTGATCTCTG |
| *Klk14* | Fw – CCTGGGCAAGCACAACATAAG |
| | Rv – CTTCAGCAGCATGAGGTCATT |
| *Gapdh* | Fw – AAGAGGGATGCTGCCCTTAC |
| | Rv – TACGGCCAAATCCGTTCACA |
| *Rn18S* | Fw – GCAATTATTCCCCATGAACG |
| | Rv – GGCCTCACTAAACCATCCAA |
| *Hprt* | Fw – CAGTCCCAGCGTCGTGATTA |
| | Rv – CACTTTTTCCAAATCCTCGGCA |
| *Ifng* | Mm99999071_m1 |
| *Aire* | Mm00477461_m1 |
| *Il22* | Mm01226722_g1 |
| *Sphk1* | Mm00448841_g1 |
| *Cd69* | Mm01183378_m1 |
| *Il17a* | Mm00439619_m1 |
| *Klk5* | Mm01203811_m1 |
| *Klk7* | Mm01197332_g1 |
| *Klk13* | Mm01197335_m1 |
| *Rn18s* | Mm03928990_g1 |
| *Gapdh* | Mm99999915_g1 |
| *Hprt* | Mm01545399_m1 |

Minus First Strand cDNA synthesis kit (ThermoFisherScientific, #K1632) and random hexamere primers. qPCR was performed using gene-specific primers and MESA Green reagent (Eurogentec) or pre-designed TaqMan gene expression assays and TaqMan Universal Master Mix II (ThermoFisherScientific) (Table 2). Samples were analyzed using CFX384 Touch Real-Time PCR Detection System (BioRad). Ct values were normalized to the mean of the Ct values of *Rn18s*, *Hrpt* and *Gapdh* reference genes. Relative expression ratios were calculated using the Pfaffl method[98].

**Antibodies and reagents**. Antibodies to the following antigens were used for flow cytometry: CD3-FITC (BD Biosciences, #555274), CD4-APC/Cy7 (BD Pharmingen, #552051), CD8a-PE (Sony Biotechnology, #1103540), CD44-Pacific Blue (Biolegend, #103020), CD25-PerCP/Cy5.5 (Sony Biotechnology, #1109560), TCRβ-PE/Cy7 (Biolegend, #109222), TCR Vδ4-BV605 (BD Biosciences, #745116), IgM-APC (Sony Biotechnology, #2632545), CD45R/B220-PerpCP/Cy5.5 (Sony Biotechnology, #1116180).

Primary antibodies used for immunofluorescence staining of antigens on paraffin sections: LEKTI (Cloud Clone Corp., #PAH145Mu01), S100A8 (Cell Signaling Technology, #47310), S100A9 (Cell Signaling Technology, #73425), KRT6A (Biolegend, #905701), LOR (Biolegend, #PRB-145P), TSLP (R&D Systems, #AF555), IL-36A (R&D Systems, #AF2297), IVL (Biolegend, #924401), CDH1 (Cell Signaling Technology, #3195), KRT10 (Biolegend, #905401 and #905403), KRT14 (Abcam, #ab181595),

KRT14 (Biolegend, #906004), KRT5 (Biolegend, #905903), Ki67 (Abcam, #ab15580), IL-24 (R&D Systems, #MAB2786), pSTAT3(Tyr705) (Cell Signaling Technology, #9145), FOXP3 (Cell Signaling Technology, #12653), IL-17A (Abcam, #ab79056), CD19 (Cell Signaling Technology, #90176), CD3 (Abcam, #ab11089), CD34 (Abcam, # ab8158), VIMENTIN (Cell Signaling Technology, #5741), KLK5 (Abbexa, #abx100977), KLK6 (Abbexa, # abx100979), KLK7 (R&D Systems, # AF2624), KLK13 (Abbexa, # abx129912) and KLK14 (Abcam, # ab278500).

Primary antibodies used for immunofluorescence staining of antigens on cryosections: CD4 (BD Pharmingen, #553727), CD3e (BD Pharmingen, #555273), CD8a (BD Pharmingen, #553027), CXCL3 (ThermoFisherScientific, #PA5-103136), F4/80 (Abcam, #ab6640), DSP (Proteintech, #25318-1-AP), FLG (Biolegend, #PRB-417), DSG1 (Santa Cruz Biotechnology, #sc20114), and Ly-6G/C (Abcam, NIMP-R14, #ab2557).

Primary antibodies used for SDS-PAGE/Western blot: LEKTI (Cloud Clone Corp., #PAH145Mu01).

Secondary antibodies used for immunofluorescence staining of tissue sections: goat anti-rat AlexaFluor555 (ThermoFisherScientific, #A21434), goat anti-rat AlexaFluor488 (ThermoFisherScientific, #A11006), goat anti-rabbit AlexaFluor555 (ThermoFisherScientific, #A21430), goat anti-chicken AlexaFluor488 (ThermoFischerScientific, # A11039), donkey anti-goat AlexaFluor546 (ThermoFisherScientific, #A11056) and donkey anti-chicken AlexaFluor488 (ThermoFischerScientific, #A78948).

Secondary antibodies used for SDS-PAGE/Western blot: goat anti-rabbit IRDye 680RD (Li-COR Biosciences, # 926-68071).

Dyes and reagents: 4′,6-diamidino-2-phenylindole, DAPI (ThermoFisherScientifc, # 62248), fixable aqua dead cell stain (ThermoFisherScientifc, # L34957), trypan blue (Bio-Rad, #1450021), fixation medium A (Fix&Perm kit, ThermoFisherScientific, # GAS001), RBC lysis buffer (eBiosciences, #00-4333), tamoxifen (Sigma, #T5648), 4-hydroxytamoxifen (Sigma, #H6278), Trizol (ThermoFisherScientific, #15596026), streptomycin sulfate salt (Sigma, #S9137), metronidazole (Sigma, #M1547), ampicillin (Sigma, #A9393), Brain and Heart Infusion agar (Sigma, #70138), mannitol salt agar (Sigma, #1.05404.0500), egg yolk emulsion (Sigma, #17148) and sucrose (Sigma, #S7903).

**Quantification of serum immunoglobulin**. Serum IgE levels were measured using mouse IgE ELISA kit (Thermo-FisherScientific, #EMIGHE). Data were analyzed in GraphPad Prism 9 software by fitting the IgE standards' raw values to a four-parameter logistic regression (4PL) model and interpolating sample IgE concentrations.

**Histological analysis**. Tissue samples were fixed in 10% neutral buffered formalin for 24 h at room temperature and subsequently embedded in paraffin. Sections of ~5 μm thickness were stained with hematoxylin and eosin. For detection of mast cells and collagen fibers in the skin, paraffin sections were stained with toluidine blue or Masson's trichrome stain, respectively. Stained sections were scanned using NanoZoomer-SQ Digital slide scanner (Hamamatsu) and images were analyzed using NDP.view2 software (Hamamatsu). Epidermal thickness was measured manually at every 0.25 μm of epidermis length and the average thickness per section was calculated. Mast cells were counted manually using NDP.view2 software. The average number of mast cells per mm$^2$ of area was determined from measurements of the entire section. For quantification of collagen fibers (blue stain signal) in Masson's trichrome stainings, snapshots of scanned whole section images were exported at 10x magnification as .tiff files using QuPath software[99]. The exported images were then analyzed in Fiji[100] to quantify the area of blue

signal as previously described[101]. The sum of the areas positive for blue stain was divided by the total dermis area in the image to obtain percent of blue-stained area for each image.

**Immunofluorescence staining, fluorescence microscopy, and image analyses**. Tissue samples were either embedded in Tissue-Tek O.C.T compound (Sakura Finetek) and kept frozen at −80 °C or fixed in 10% neutral buffered formalin for 24 h at room temperature and then embedded in paraffin. For immunofluorescence staining of DSG1, DSP, FLG, CXCL3, CD4, CD3 or CD8, cryosections (10 μm thickness) were fixed in 4% paraformaldehyde and permeabilized in ice-cold acetone. Sections were then blocked in PBS buffer containing 5% bovine serum albumin and incubated with primary antibody overnight at 4 °C (CXCL3, CD4, CD3 and CD8) or for 1 h at room temperature (DSG1, DSP and FLG). Sections were then washed and incubated with fluorophore-labeled secondary antibody. Finally, sections were washed, counterstained with DAPI and mounted with ProLong Gold antifade medium (ThermoFisherScientific).

Immunofluorescence stainings of LEKTI, S100A8, S100A9, KRT6A, LOR, IVL, CDH1, KRT10, KRT14, Ki67, TSLP, IL-36A, IL-24, pSTAT3, FOXP3, IL-17A as wells as the double immunofluorescence stainings of CD34/VIM, KRT10/KRT14, KLK5/KRT5, KLK6/KRT5, KLK7/KRT5, KLK13/KRT5 and KLK14/KRT5 were performed on paraffin sections (5 μm thickness). Antigen retrieval was performed by heating slides in citrate buffer (10 mM sodium citrate, pH6.0, 0.05% Tween-20) at 96 °C for 45 min. For staining of pSTAT3, antigen retrieval was done by heating slides in EDTA 1 mM pH8.0 buffer at 96 °C for 45 min. Sections were washed in PBS, blocked for 30 min in blocking solution (PBS, 5% BSA, 0.05% Tween-20), and incubated in primary antibody solution overnight at 4 °C (S100A8, S100A9, KRT6A, IVL, CDH1, KRT10, KRT14, Ki67, TSLP, IL-36A, IL-24, pSTAT3, FOXP3, IL-17A, and all double immunofluorescence stainings) or for 1 h at room temperature (LEKTI, LOR). Sections were then washed, incubated with fluorophore-labeled secondary antibody and washed again. Sections were counterstained with DAPI and mounted with ProLong Gold antifade medium (ThermoFisherScientific).

Stained sections were imaged using Leica TCS SP8 SMD confocal microscope (Leica Microsystems) with 40x objective. Images were processed using Fiji. For visualization, images represent maximum projections of a Z-stack or a single slice in the case of double immunostaining. For quantification of fluorescence intensity signal, the average projection of a Z-stack with an equal number of slices for each image was used. In each image, the epidermis and two regions outside the tissue section (used for background subtraction) were selected and fluorescence intensity values and area were measured. Quantification of stained immune cell infiltrates in the dermis or the number of KLK+ cells in thymic medulla was performed manually in Fiji by marking every positive cell (point tool) and measuring the skin dermis area or the thymic medullary epithelium area. For quantification of CD34+ and CD34+VIM+ cells, VIM+ cells were selected manually in Fiji (rectangle tool) and then the mean fluorescence intensities in the green (VIM) and red (CD34) channels were independently measured for each cell. Threshold mean signal intensity values that define VIM+ or CD34+ cells were set in images of WT control skin samples and then used to calculate the number of VIM+ and VIM+CD34+ cells per dermis area in all images using a custom script in Python. At least three images per skin section were analyzed (one skin section per mouse). For quantification of KRT10 and KRT14 immunofluorescence signal co-colocalization, the epidermis region in single slice images from Z-stacks was selected. Green (KRT14) and red

(KRT10) channels were then split and the co-localization of staining signal was calculated only in the selected region (epidermis) using Coloc 2 tool in Fiji. Pearson correlation coefficient values were plotted.

**Measurement of LEKTI protein levels in skin by SDS-PAGE/Western blot**. Flash-frozen skin samples were crushed to powder using cellcrusher instrument (Cellcrusher). The powder was lysed in RIPA lysis buffer (ThermoFischerScientific, #89901) supplemented with cOmplete protease inhibitor cocktail (Sigma, #11836170001) by vortexing and incubation on ice for 10 min. Lysates were centrifuged at 14 000 rpm and 4 °C for 10 min. The total protein concentration in the cleared supernatants was determined using Pierce BCA protein assay kit (ThermoFischerScientific, #23225). Protein samples were mixed with 4x NuPAGE LDS sample buffer (ThermoFischerScientific, #NP0007) and 2-mercaptoethanol (Sigma, #63689) to 2% final concentration and boiled at 70 °C for 10 min. Protein samples (50 µg total protein per lane) were then resolved on a 4–12% Bis-Tris gel (ThermoFischerScientific, #NP0321BOX) in MOPS-SDS running buffer (ThermoFischerScientific, #NP0001). The resolved proteins were transferred to a 0.22 µm nitrocellulose membrane by liquid transfer using Tris/Glycine buffer and BioRad Mini Trans-Blot Cell for 3 h at 80 V and 160 mA. Total proteins on the membrane were stained by incubation in Ponceau S solution (Sigma, #P7170) for 3–5 min. The membrane was imaged and then incubated in blocking buffer (5% w/v nonfat dry milk, 0.1% Tween-20 in PBS) for 30 min. The membrane was incubated with primary antibody diluted in blocking buffer (1/1000 dilution) overnight at 4 °C. After three washes for 10 min each in 0.1% Tween-20 PBS buffer, the membrane was incubated with secondary antibody diluted in blocking buffer (1/10 000 dilution) for 1 h at room temperature. After final washes, the membrane was imaged using Odyssey CLx near-infrared fluorescence imaging system (Li-COR Biosciences). Lanes in the images were quantified using Image Studio software (Li-COR Biosciences) and Fiji. Protein levels in bands of interest were normalized to total protein on the membrane (Ponceau S stain) and expressed as ratio to average of protein levels in the samples from WT control mice.

**Measurement of protease activity in skin and thymus extracts**. Mouse back skin or thymus tissue samples were homogenized in cold 1 M acetic acid using Qiagen Tissue Lyser LT instrument and 5 mm stainless steel beads (Qiagen, #69989). Homogenates were incubated at 4 °C overnight. Extracts were then cleared by centrifugation at 4 °C and 13,000×g for 30 min. Cleared supernatants were spun in Savant SpeedVac SPD111V instrument (ThermoFischerScientific) at 35 °C until dryness. Dried protein extracts were resuspended in water by overnight incubation at 4 °C. The resulting water protein extracts were cleared by centrifugation at 4 °C and 13 000xg for 10 min. Total protein concentration was determined using QuantiPro™ BCA Assay Kit (Sigma). Protein samples were diluted to same concentration in water and equal volume of diluted protein extract (corresponding to 5 µg total protein for skin extracts or 2 µg total protein for thymus extracts) was distributed in duplicates in black, non-binding 96-well microplates (Greiner bio-one). A mixture of activity buffer (100 mM Tris HCl, 100 mM NaCl, 0.005% Triton X-100 pH8.0) and fluorogenic substrate for trypsin-like serine proteases Boc-VPR-amc (Bachem), KLK7-preferred substrate KHLY-amc (custom synthesis by Proteogenix, France), or KLK14-preferred substrate Ac-WAVR-amc (custom synthesis by Proteogenix, France) were added to each well to reach a final substrate concentration of 100 µM. Samples for background subtraction (blank) contained water, assay buffer and substrate. For protease

inhibition, protein extracts were pre-incubated in activity buffer containing GSK951A[36] or DMSO (control), KLK7-specific inhibitor pepPG278 or control inactive inhibitor pepPG303[39] at 37 °C for 1 h before addition of substrate. The final assay concentration of inhibitor was 10 µM (skin extracts) or 1 µM (thymus extracts). After addition of substrate, the plates were incubated at 37 °C for 24 h. Fluorescence intensity was measured at room temperature 24 h after addition of substrate using EnVision2105 Multimode plate reader (Perkin Elmer) with excitation filter 355 nm and emission filter 460 nm. Data were analyzed in Excel and visualized in GraphPad Prism 9 software. For background noise subtraction, the average of fluorescence intensity read counts (RFU values) of blank samples (background control) was subtracted from the average of RFU values of each protein extract sample. Samples that had negative RFU values after blank subtraction were considered samples with no detected protease activity and therefore were assigned a value of 0.

**In vitro recombinant protein digestion assay**. Proteolytic digestion of human full-length AHSG, human full-length IL-36A or human full-length IL-36G recombinant proteins was performed in activity buffer (100 mM Tris HCl, 100 mM NaCl, 0.005% Triton X-100 pH8.0) at enzyme to substrate molar ratios of 1:25, 1:50 and 1:100. Recombinant AHSG (100 ng, 80 nM), IL-36A (48 ng, 160 nM) or IL-36G (48 ng, 160 nM) was mixed with recombinant human enzyme (KLK or TMPRSS11D at final assay concentration of 1.6 nM, 3.2 nM, or 6.4 nM corresponding to 0.7 ng, 1.45 ng and 2.9 ng respectively) and incubated at 37 °C. Digestion reactions were stopped at different time points (5 min, 15 min, 30 min, 1 h, 2 h or 24 h) by adding reducing agent and LDS sample buffer and boiling the reaction for 10 min at 70 °C. Digestion reactions without enzyme or without substrate were incubated in parallel at 37 °C for 2 or 24 h and served as negative controls. The digestion reactions were analyzed by 4–12% gradient SDS-polyacrylamide gel electrophoresis and silver staining using Pierce Silver stain kit (ThermoFisherScientific, #24612). Recombinant KLK7, KLK8, KLK13 and KLK14 pro-enzymes were activated by incubation with bacterial thermolysin for 2 h at 37 °C. Recombinant KLK6 pro-enzyme was activated by incubation for 2 h at room temperature with bacterial thermolysin at 1:110 thermolysin/KLK6 molar ratio. Recombinant KLK13 pro-enzyme was self-activated by incubation in activity buffer for 24 h at 37 °C. The activity of all enzymes was verified by fluorogenic peptide substrate cleavage assay in the same activity buffer at final concentration of enzyme and substrate equal to 1.6 nM and 100 µM, respectively (Supplementary Fig. 12). All recombinant proteins or enzymes were purchased from R&D Systems.

**Blood analyses**. Blood from 5- to 8-week-old mice was collected into EDTA K3E microtubes (Sarstedt) by cardiac puncture immediately after euthanasia and analyzed using ProCyte Dx® Hematology Analyzer (Idexx).

**Flow cytometry**. Lymphoid organs were collected from 4-week-old *Spink5* cKO mice (*KRT14-CreERT2*[(Tg/0)]/*Spink5*[fl/fl] and/or *KRT14-CreERT2*[(Tg/0)]/*Spink5*[fl/-]) and control littermate mice (*Spink5*[fl/fl] and/or *Spink5*[fl/-]). Single-cell suspensions from the whole thymus, spleen, and two inguinal lymph nodes per mouse were obtained by grinding the tissue with a syringe plunger against 70 µm mesh of a cell strainer in cold staining buffer (2 mM EDTA PBS buffer supplemented with 5% fetal bovine serum). Cell number and viability were determined using trypan blue dye and TC20 automated cell counter (Bio-Rad). Cells (1 million viable cells) were re-suspended in staining buffer containing a mix of fixable aqua dead cell stain

(ThermoFisherScientifc, #L34957) and fluorescently-labeled antibodies for detection of B cells (anti-IgM, anti-B220) and T cells (anti-CD3, anti-CD4, anti-CD8) in lymph nodes and spleens, or thymocytes (anti-TCRβ, anti-TCRδ, anti-CD44, anti-CD25) and incubated for 30 min at 4 °C. Cells were washed once in cold staining buffer and then fixed with medium A reagent (Fix&Perm kit, ThermoFisherScientific, #GAS001). Fixed cells were washed, re-suspended in staining buffer and analyzed using LSRFortessa X-20 flow cytometer (BD Biosciences). Data were analyzed using FlowJo™ v10.8 Software (BD Biosciences).

**Analysis of thymic TCRα repertoire**. Thymocytes were isolated from thymi of five *Spink5* cKO and five wild-type littermate mice at an average age of 5 weeks. Total RNA from thymocytes ($10 \times 10^6$ live cells) was extracted using Trizol reagent (Thermo-FisherScientific). Synthesis and amplification of cDNA by 5'-RACE RT-PCR and subsequent deep sequencing and data analyses were performed as described previously[102].

**Detection and quantification of bacteria in skin and internal organs**. Skin swabs were collected from shaved back skin of four *Spink5* cKO mice and two control mice on the same day. *Staphylococcus aureus* (*S.aureus*) was detected by quantitative PCR according to procedures and protocols at Charles River Laboratories. The estimated DNA copy number for each *S.aureus* assay was reported. For quantification of bacterial load in skin, blood and internal organs (spleen, liver, lungs) of WT control and *Spink5* cKO mice, tissues were mixed with 0.5 mL of sterile PBS in 2 mL lysing matrix D tubes each containing one ¼" ceramic sphere (MP Biomedicals). Tissues were homogenized using FastPrep-24 tissue homogenizer (MP Biomedicals) for 3 shake cycles of 20 seconds each. The homogenates were spread on pre-warmed Brain and Heart Infusion agar and mannitol salt agar plates (100 µL of homogenate per plate). Skin homogenates were diluted 1/50 in sterile PBS before plating. Plates were incubated at 37 °C in aerobic conditions for 48 h. Colonies were counted and CFUs/g values were calculated considering the weight of the harvested tissue samples, the volume of homogenate spread per plate and the dilution factor (in case of skin samples). Mannitol salt agar plates contained 5% v/v egg yolk emulsion.

**Statistics and reproducibility**. Statistical analyses were performed with GraphPad Prism v9.1.2 (GraphPad Software) or with custom Python script for Fisher's r-to-z and standardized residuals tests. *P*-values of statistical significance tests for experiments comparing 2 groups were calculated using either two-tailed non-parametric Wilcoxon matched-pairs signed rank test or two-tailed unpaired non-parametric Mann–Whitney test. Survival analyses were performed using log rank test. *P*-values lower than 0.05 were considered statistically significant. Experiments were performed with tissue samples from at least 5 different mice per experimental group unless otherwise indicated.

**Reporting summary**. Further information on research design is available in the Nature Portfolio Reporting Summary linked to this article.

## Data availability

The transcriptomic data were deposited at the Gene Expression Omnibus (GEO) data repository under accession number GSE224280 (*Spink5* cKO skin transcriptome), GSE224409 (*Spink5* cKO lymph node transcriptome), GSE224673 (Tg.*hKLK5* skin transcriptome), and GSE224674 (*Spink5*⁻/⁻ skin transcriptome). Mass spectrometry data (raw spectral files and intermediary identification search results) were uploaded to the MassIVE repository (part of the ProteomeXchange consortium). Data can be accessed using the MassIVE accession number MSV000091184 or the ProteomeXchange

accession number PXD039796. Source data for all graphs in this study are provided in Supplementary Data 6 of the paper. Images of uncropped and unedited gels/blots are provided in Supplementary Figs. 20–23.

## Code availability

Custom code (Python scripts) used for data analyses in this study are available upon request.

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

## Acknowledgements

We are thankful to the animal facility as well as the genomics, imaging and histology platforms at Imagine Institute for excellent technical assistance. We thank Z. Allouche, L. Motreff, L. Lemée, C. Proux, B. Jagla, and J.Y. Coppée at the Biomics Platform, C2RT, Institut Pasteur, Paris, France, supported by France Génomique (ANR-10-INBS-09-09) and IBISA for RNA sequencing of tissue samples from mouse models. We are grateful to Christian Heinis for providing the KLK7-specific inhibitor pepPG278 and the control inhibitor pepPG303. We would like to acknowledge the funding supports from Agence Nationale de la Recherche grants PAR2-PATH: ANR-15-CE14-0009, KLKIN: ANR-15-RAR3-0005-01, TARGET-NS: ANR-19-CE17-0017-02, Tfh-Atopy: ANR-17-CE14-0025 and Association Ichthyose France. The funders had no role in study design, data collection and analysis, decision to publish, or preparation of the manuscript.

## Author contributions

E.P., J.M.L.G., M.F., J.P.V., O.S. and A.H. designed the study; E.P., M.F., F.L., C.B. and J.P.V. performed the experiments; P.G. performed the development, synthesis, purification, and characterization of the KLK7-specific inhibitor pepPG278 and the control inactive inhibitor pepPG303; E.P., J.M.L.G., M.F., J.P.V. and L.C.T. analyzed the data; E.P. and J.M.L.G. wrote the manuscript; J.M.L.G., M.F., J.P.V., O.S., J.E.G. and A.H. revised the manuscript.

## Competing interests

The authors declare the following competing interest(s): E.P., C.B. and A.H. are inventors of a patent. The rest of the authors declare no competing interests.
