## [Peer Review File · Communications Biology]

Reviewers' comments:

Reviewer #1 (Remarks to the Author):

This is an interesting and high-quality research manuscript from Hovnanian group who has made seminar contribution to this field. In this manuscript, they performed a comprehensive comparison of Netherton syndrome, SPINK5 cKO, and hKLK5 Tg mice skin tissues to understand the disease pathogenesis. Proteomics and transcripts analysis between Spink5cKO and NS skin samples reveals dysregulated immune infiltrations, protease activation, and dermatological manifestations. The study provides insight on the NS pathogenesis and provides a preclinical model for testing the treatment of NS, which is a severe Ichthyosis with limited treatment options. The manuscript is well-written and overall study design and interpretation are sound.

We have two suggestions for further improvement.

1. While the characterization of skin biopsies in SPINK5 cKO is clear and convincing, the investigation of thymus and spleen innate and adaptive immunity is limited, and interpretation may not be correct. Given the extensive immune activation and skin lesions in these animals, thymic atrophy could simply reflect stress and stress induced thymus atrophy, especially "the magnitude of DP thymocyte depletion in Spink5 cKO mice strongly correlated with the severity of their skin phenotype".

While it is difficult to investigate the mechanism of thymus atrophy without additional genetic tool development (e.g., thymus specific Spink5KO), the authors have unique tools and reagents to quantify the protease activity in the thymic tissues. Are KLK5, KLK7, KLK14, or any SPINK5 related protease expressed in the thymus medullary epithelial cells? Are tryptic or chymotryptic activity elevated in thymic tissues? Such analysis would substantiate the hypothesis on the role of SPINK5 in thymus.

2. The skin swab and detection of "S. aureus" cannot support "the Spink5 cKO could be suffering from systemic bacterial infection" (line465). Up to 30% healthy individuals have S. aureus in the skin/nose. To support "systemic bacterial infection" (lines 465, 471, 512, 532 and more replaces), the author should quantify S. aureus or other bacteria in internal organs (e.g., lung, liver) and/or perform experiment to with pan-antibiotic treatment in Spink5 cKO mice to test whether it may rescue the phenotype of neutrophil in circulation. Such experiments will be extremely valuable to support such statement/hypothesis brought up many times in the manuscript. As an alternative hypothesis of systemic infection to increase neutrophils in spleen and blood, trypsin-like protease (PMID 9550425) and even SPINK5 regulated KLK5 (PMID 35487308) have been reported to induce sterile neutrophilic inflammation.

Reviewer #2 (Remarks to the Author):

Petrova E et al. investigated the uncovered pathogenesis of Netherton syndrome (NS) using samples from patients and Spink5 conditional knockout mice (cKO). The authors compared the differences between Spink5 cKO and previously used NS model mice, such as Tg.hKLK5 mice and Spink5^{-/-} mice, and showed that Spink5 cKO mice are superior model mice for human NS. They successfully demonstrated the importance of up-regulated gene profiles regarding "protease activity," such as KLKs and SERPINS, and "immune and inflammation response," such as IL-17, IL-36, and IL-20 families by comparative transcriptome and proteomics analysis using Spink5 cKO mice and NS patients. Furthermore, they identified that KLK14 and KLK7 could modify the inflammation signaling via AHSG

and IL-36 A and IL-36G processing, respectively. Finally, they analyzed the relationship between systemic inflammation and immune abnormalities in the thymus, lymph node, and spleen. Their findings could contribute to a better understanding of NS pathogenesis and the development of effective treatments. Overall, the methods used in this study were deemed appropriate, and the results and discussion were convincing in explaining the conclusions. Specific comments are shown below.

Major point

1. On page 4, line 142, the authors stated that "the severity of the skin lesion strongly correlated with the absence of LEKTI protein," supported by immunofluorescence images shown in Supplementary Fig2h. If there are additional quantitative data, such as Western blot results, please include them. And, why does it appear that the expression of LEKTI is decreased more in the lesional area than in the non-lesional area in Spink5 cKO mice?
2. Was double staining for KRT14 and KRT10 conducted in Figure 2n? If the tissue section was cut at an angle, it is possible that the basal layer appears thicker, which could exaggerate the observed increase in KRT14 expression.
3. On page 8, line 310, it is mentioned that CD34 is the marker of hair follicle stem cells. However, it is also one of the markers associated with fibroblasts. As a result, could the decrease in CD34-positive fibroblasts potentially lead to reducing collagen fibers?

Minor point

4. Have you ever checked if pepPG278 can alleviate skin inflammation in Spink5 cKO mice?
5. In Fig 3d, the white text with a yellow background in the center of the Venn diagram is hard to read.
6. On page 11, line 420, which Figure shows the correlation between Spleen size and phenotype severity?
7. Isn't Figure 7g correct for Figure 7f on page 12, line 489?

Reviewer #3 (Remarks to the Author):

The manuscript describes a novel Spink5 conditional knockout (cKO) mouse model for Netherton syndrome (NS), a severe and potentially lethal rare genetic skin disease. The interesting aspect of this model is the survival of the animals for several weeks, which should enable the investigation of new therapeutic approaches. This is a substantial improvement over previously described Spink5 knockout models (perinatally lethal), skin grafting of neonatal Spink5^{-/-} skin onto normal mice (requiring the sacrifice of many animals) or knockout models of individual proteases (representing only partial aspects of the human disease, since intact LEKTI is still present).

The authors present an in-depth characterization of the model on the molecular, histologic, enzyme and functional levels and show a good correlation between the transcriptomes and proteomes of their knockout mice and the human disease. Therefore, the model will be of interest to other investigators in the field and may promote the identification of novel treatments.

The manuscript is a bit difficult to read in many places, because it is very long and the reader has to

switch between so many parts of Figures and Supplementary Figures. It should be possible to reduce the information in the supplements or indicate the important information in individual gel graphs, e.g., by arrows. Some details about the comparison of Spink5 cKO and Tg.hKLK5 mice could also be shifted to the supplementary information or even, for the sake of clarity, be omitted from this paper, since the main goal is to show the validity of the new model for human NS.

The description of the experimental procedures is very detailed and should allow reproduction.

In the following, I will address individual aspects, which could be improved or should be changed.

Introduction:

- The authors should reference Guo et al, 2022: Establishment of a mouse model of Netherton syndrome based on CRISPR/Cas9 technology, *Eur J Dermatol.* 2022 Jul 1;32(4):459-463. doi: 10.1684/ejd.2022.4287, and explain what differentiates their model from the one described by Guo et al.

Results:

- Lines 118 – 125: Consider changing the order of spontaneous floxed allele excision \diamond induced deletion \diamond spontaneous floxed allele excision to describing first one and then the other.
- Lines 141 – 145: While the lack of correlation between Spink5 mRNA expression and skin severity score (Fig. S2i) is clear, Fig. S2h does not prove a correlation between the number of LEKTI-positive cells and phenotype severity, since only one mouse is shown.
- Line 161: "(Figure 2b)" should read (Figure 2a).
- Line 181: should read ... Supplementary Figure S4d-e.
- Line 193: should read ...Supplementary Figure S4h-k'.
- Lines 193 – 196: I cannot follow the author's argumentation that the overall expression of FLG, DSG1 and DSP were unchanged. According to Suppl. Fig. S4j', DSG1 integrated fluorescence intensity is even increased, as I would assume from Fig.2i. This is rather surprising, considering that DSG1 is a KLK substrate and reduced in NS skin. The authors should comment on the increased staining of FLG, DSG1 and DSP in Figures 2h – j.
- Line 283: "... which is driven by IL-17, IL-36 and IL-20 family members" is formulated a bit too strongly. Just because the mRNAs for these cytokines are upregulated does not necessarily imply a causative role for the disease process.
- Lines 339 – 341: it would be sufficient to show either Fig. 5e or Fig 5f (5f preferred for the sake of comparison with 5g and 5h).
- Lines 341 and 343: Suppl. Fig. 12 does not contain information about AHSG and should, therefore, be quoted differently.
- Line 357: Suppl. Fig. 11l does not provide any information about cleavage of IL-36A, in fact, it is redundant with S11d-k and could be left out.
- Lines 357 – 359: "IL-36G, another member of the IL-36 pro-inflammatory cytokine family that is highly upregulated at the mRNA and protein levels in skin of NS patients...". This is in direct contrast to line 351, "IL-36A protein could not be detected in NS patients...". Please clarify!
- Line 365: "...proteomics analyses confirm IL-36 and IL-17 signaling...". Since IL-17 has not been mentioned in the proteomics results section so far, this statement comes rather as a surprise. Please explain!
- Lines 382 – 383: the distribution of graphs between Fig. 6 and S13 is difficult to understand. Why are some markers quantified in Fig 6, but not others, which would be equally important (e.g., Fig. 6g-k, Fig. S13h)?
- Line 391: IL-36A ... was strongly upregulated in epidermal keratinocytes: how does this fit in with IL-36A being degraded by kallikreins?
- Line 420: Fig. S14c and d do not show a correlation between thymus size and severity.
- Lines 422-423: Fig. 7f-g seem to be identical to Suppl. Fig. S15a and b, except for the label of the y-axis. Please clarify!

- Line 427: The SP thymocytes are shown in Fig. 7g, not 7f.
- Line 464: please explain how the Spink5 cKO mice could have been infected with *S. aureus*, since they were kept in a specific pathogen-free facility in IVC cages. Where does the infection come in?
- Lines 488-489: "Fig 7f" should read S18f. The fold change of 67 cannot be deduced from this graph.
- Lines 490 – 494: It is not clear what data they relate to.

Discussion:

The discussion is comprehensive, well balanced and presents points for further investigation.

Materials and Methods:

This section is very detailed and should allow other investigators to replicate the results.

- Lines 679 – 682: The authors may want to explain why they used either i.p.- or topical induction of the knockout phenotype and whether there were any differences in outcome.

Figure Legends:

- Line 1545: If the organ weight index were calculated as described, the values should be <1 . Please replace by the explanation in line 1900.
- Lines 1643 – 1644: Instead of saying "...from a Spink5 cKO mouse", the authors may want to point out that this is the mouse of Fig. 2d.
- Lines 1782 – 1784: It would be helpful for the reader to mention not only Fig. S10, but also Fig. S9d-e.

Figures:

- Figure 5: Fig. 5d could be moved to the supplements in order to focus more on the comparison Spink5 cKO to human NS in the main part of the document. Fig. 5e may not be necessary (cf. above).
- Suppl. Fig. 17a-d: What does the abbreviation dLN stand for? The figure legend mentions inguinal lymph nodes.

Reviewers' comments:

Reviewer #1 (Remarks to the Author):

This is an interesting and high-quality research manuscript from Hovnanian group who has made seminar contribution to this field. In this manuscript, they performed a comprehensive comparison of Netherton syndrome, SPINK5 cKO, and hKLK5 Tg mice skin tissues to understand the disease pathogenesis. Proteomics and transcripts analysis between Spink5cKO and NS skin samples reveals dysregulated immune infiltrations, protease activation, and dermatological manifestations. The study provides insight on the NS pathogenesis and provides a preclinical model for testing the treatment of NS, which is a severe Ichthyosis with limited treatment options. The manuscript is well-written and overall study design and interpretation are sound.

We have two suggestions for further improvement.

1. While the characterization of skin biopsies in SPINK5 cKO is clear and convincing, the investigation of thymus and spleen innate and adaptive immunity is limited, and interpretation may not be correct. Given the extensive immune activation and skin lesions in these animals, thymic atrophy could simply reflect stress and stress induced thymus atrophy, especially “the magnitude of DP thymocyte depletion in Spink5 cKO mice strongly correlated with the severity of their skin phenotype”.

While it is difficult to investigate the mechanism of thymus atrophy without additional genetic tool development (e.g., thymus specific Spink5KO), the authors have unique tools and reagents to quantify the protease activity in the thymic tissues. Are KLK5, KLK7, KLK14, or any SPINK5 related protease expressed in the thymus medullary epithelial cells? Are tryptic or chymotryptic activity elevated in thymic tissues? Such analysis would substantialize the hypothesis on the role of SPINK5 in thymus.

We thank reviewer #1 for the comments and suggestions to study the role of Spink5 in the thymus and its possible connection with thymic atrophy. We agree with reviewer #1 that thymic atrophy in *Spink5* cKO mice could be a result of stress due to the severe skin and systemic inflammation and we have modified the manuscript accordingly (lines 559-560, 574-578). We have performed additional experiments and have included the results in the manuscript and figures (lines 559-662, Figure 8 and Suppl. Fig. 19).

As suggested by reviewer #1, we measured the expression of several KLKs (at both mRNA and protein levels) and the tryptic and chymotryptic protease activities in thymus tissue of *Spink5* cKO mice and WT control mice.

A) We measured the mRNA expression level of mouse *Klk5*, *Klk6*, *Klk7*, *Klk8*, *Klk13* and *Klk14* in thymus tissue using RT-qPCR. At the mRNA level, the expression levels of mouse *Klk5*, *Klk6*, *Klk7*, *Klk8*, *Klk13* and *Klk14* are slightly higher in *Spink5* cKO thymus as compared to controls (Suppl. Fig.19a-f, *Klk5* 1.6-fold increase, *Klk6* 1.8-fold, *Klk7* 1.7-fold, *Klk8* 1.4-fold, *Klk13* 2-fold, *Klk14* 1.9-fold), however the difference is not statistically significant.

B) In addition, we analyzed the expression level and pattern of mouse KLK5, KLK6, KLK7, KLK13 and KLK14 by immunofluorescence staining of thymus paraffin sections and confocal fluorescence microscopy imaging of stained sections. At the protein level, mouse KLK5, KLK6, KLK7, KLK13 and KLK14

were expressed in distinct epithelial cells of the thymic medullary epithelium. This expression pattern was similar in *Spink5* cKO and control thymi (Fig.8a-c', Suppl. Fig.19g-h'). The number of KLK-positive cells is slightly increased in the medullary thymic epithelium of *Spink5* cKO mice as compared to WT controls (Fig.8a-c', Suppl. Fig.19g-h', KLK5 1.3-fold increase, KLK6 1.2-fold, KLK7 1.4-fold, KLK13 1.5-fold, KLK14 1.1-fold). However, this slight difference is not statistically significant.

Our findings on mouse KLK expression pattern in the thymic medulla is in agreement with previous single-cell transcriptome analyses of mouse medullary thymic epithelial cells ¹.

C) Next, we measured tryptic and chymotryptic protease activities in thymus tissue extracts using the same fluorogenic substrate cleavage assay as the one that we used for measuring protease activity in skin extracts (Figure 2a-c). We observed a 2.4-fold increase of VPR broad-spectrum substrate cleavage signal (KLK5-like activity), 1.6-fold increase of WAVR substrate cleavage signal (KLK14-like activity) and 21-fold increase of KHLV cleavage signal (KLK7-like activity) in *Spink5* cKO thymus extracts as compared to WT control thymus extracts (Fig. 8d-e). Although the fluorescence signal resulting from substrate cleavage was much lower as compared to the one observed in skin extracts, the difference between *Spink5* cKO and WT groups was statistically significant. Moreover, co-incubation of the thymus tissue extracts with KLK5- or KLK7-specific inhibitors led to 80% reduction of VPR cleavage signal (KLK5 inhibition) and complete reduction of KHLV substrate cleavage signal (KLK7 inhibition) (Fig.8d-e). Thus, given the presence of KLK5, KLK7 and KLK14 expression in medullary thymic epithelium and the sensitivity of thymic extract protease activity to KLK5- and KLK7-specific inhibitors, we conclude that KLKs can contribute to the increase of protease activity in *Spink5* cKO thymus. The protease activity values that we measured in thymus tissue extracts of WT control mice are close to background noise and, in some samples, activity could not be detected, therefore these results should be interpreted with caution and should be confirmed with more sensitive substrates or activity-based probes when these tools become available.

In conclusion, we (i) confirmed that KLKs are expressed in the thymic medullary epithelial cells (Fig. 8a-c', Suppl. Fig. 19g-h'), (ii) we found that KLKs expression levels are similar in *Spink5* cKO mice and control mice (Fig. 8a-c', Suppl. Fig. 19a-h'), and (iii) we found that trypsin-like and chymotrypsin-like serine protease activities are significantly increased in thymus tissue extracts of *Spink5* cKO mice as compared to WT control littermates, which agrees with the deletion of *Spink5* in the thymus and its role as an inhibitor of serine protease activity in the thymic epithelium. However, as mentioned by reviewer #1, further analyses with double knockout mice and thymus-specific deletion of KLKs are needed to define the role of KLKs and SPINK5 in the thymic atrophy phenotype of *Spink5* cKO mice.

2. The skin swab and detection of "S. aureus" cannot support "the *Spink5* cKO could be suffering from systemic bacterial infection" (line465). Up to 30% healthy individuals have S. aureus in the skin/nose. To support "systemic bacterial infection" (lines 465, 471, 512, 532 and more replaces), the author should quantify S. aureus or other bacteria in internal organs (e.g., lung, liver) and/or perform experiment to with pan-antibiotic treatment in *Spink5* cKO mice to test whether it may rescue the phenotype of neutrophil in circulation. Such experiments will be extremely valuable to support such statement/hypothesis brought up many times in the manuscript. As an alternative hypothesis of systemic infection to increase neutrophils in spleen and blood, trypsin-like protease (PMID 9550425) and even SPINK5 regulated KLK5 (PMID 35487308) have been reported to induce sterile neutrophilic inflammation.

We agree with the reviewer's comment; therefore, we performed the experiments suggested by the reviewer and modified the manuscript and the corresponding figures.

We quantified bacteria by plating tissue homogenates (from skin, blood, lung, liver and spleen) on Brain and Heart Infusion (BHI) agar, which is a non-selective, nutrient-rich medium permissive to the growth of many types of microorganisms including bacteria, yeasts, and molds. In addition, to detect *S.aureus* in these same tissues, we plated homogenates on mannitol salt agar plates supplemented with 5% v/v egg yolk emulsion.

We did not detect significant differences in the number of colony forming units (CFUs) per tissue weight in the organs analyzed between *Spink5* cKO and control WT mice. These results reject our initial hypothesis that systemic bacterial infection in *Spink5* cKO mice is a cause for systemic inflammation (Suppl. Fig. 18b-c). The bacteria that we detected in internal organs, such as liver and lungs, could represent microbiota as previously described ^{2,3}. The skin of *Spink5* cKO mice tends to have more bacteria in general (BHI agar cultures, Suppl. Fig. 18b) and *S.aureus* (mannitol salt agar, Suppl. Fig.18c) as compared to WT skin, but the difference is not statistically significant. The increased copies of *S.aureus* genome that we detected in *Spink5* cKO skin swabs by qPCR (Suppl. Fig.16 of the initial submission, Suppl. Fig.18a of the revised manuscript) could result from the accumulation of dead bacteria on the skin lesions with time and thus may not correlate with the quantity of viable bacteria in the skin or an active skin infection, as suggested by previous studies of skin microbiome ⁴.

To test any possible synergistic effect between bacterial infection and SPINK5 deficiency that contributes to systemic inflammation, we treated *Spink5* cKO and WT mice with broad-spectrum antibiotics cocktail (ampicillin 0.5 g/L, streptomycin 1 g/L and metronidazole 0.5 g/L) in 2% sucrose-containing drinking water for a duration of 2 weeks. The placebo groups received 2% sucrose in drinking water (Suppl. Fig. 18d-e). Antibiotics treatment was started at the same time as the induction of CreERT2 by tamoxifen. At the end of the treatment protocol, skin and internal organs (spleen, liver and lungs) were harvested for measurement of bacterial load by culture on BHI agar and mannitol salt agar. In addition, blood was analyzed for presence of neutrophils and thymus tissue was harvested for measurement of organ weight and protease activity. Despite eliminating bacteria in skin and internal organs (lung, liver, spleen) (Suppl. Fig. 18f-g), antibiotics did not prevent the appearance of skin barrier defect, skin and systemic inflammation, as evidenced by severe lesions, high transepidermal water loss, high number of neutrophils in blood and severe thymic atrophy (Suppl. Fig. 18e, h-k). Finally, to assess a possible synergistic effect between bacteria and SPINK5 deficiency, we measured serine protease activity in thymus tissue extracts from antibiotics- and placebo-treated *Spink5* cKO mice and their respective WT controls (Suppl. Fig.19i-j). Both antibiotics- and placebo-treated *Spink5* cKO mice showed similar increase in trypsin-like and chymotrypsin-like serine protease activity in thymus tissue as compared to the corresponding WT control mice (Suppl. Fig.19i-j), which correlated significantly with thymic atrophy (Suppl. Fig.19k-l).

Taken together, these results exclude systemic bacterial infection as a cause of the systemic inflammation phenotype (increased blood neutrophil counts) and the resulting thymic atrophy in *Spink5* cKO mice. As suggested by reviewer #1, the increased levels of blood and spleen neutrophils in *Spink5* cKO mice could represent sterile neutrophilic inflammation caused by increased serine protease activity. Since KLKs are known to proteolytically activate/modulate pro-inflammatory mediators such as C3 ⁵, IL-1b ⁶, CAMP ⁷ or IL-36 that we uncovered here (Figure 5), it is possible that KLK activity in *Spink5* cKO skin and/or thymus induces sterile neutrophilic inflammation. Future studies should address the molecular and cellular mechanisms behind this possible scenario.

We thank the reviewer for his/her suggestions, as the results of these experiments change some of our initial interpretations and thus improve the quality and precision of the manuscript. Accordingly, we have modified the main text of the manuscript referring to the analyses of lymphoid organs (lines 409-536 and 613-646 of the initial manuscript) and Figure 7, and we have added one more main figure

(Figure 8) and have re-organized the order of the corresponding supplementary figures, adding one more supplementary figure (Suppl. Fig. 19).

Reviewer #2 (Remarks to the Author):

Petrova E et al. investigated the uncovered pathogenesis of Netherton syndrome (NS) using samples from patients and *Spink5* conditional knockout mice (cKO). The authors compared the differences between *Spink5* cKO and previously used NS model mice, such as Tg.hKLK5 mice and *Spink5*^{-/-} mice, and showed that *Spink5* cKO mice are superior model mice for human NS. They successfully demonstrated the importance of up-regulated gene profiles regarding "protease activity," such as KLKs and SERPINs, and "immune and inflammation response," such as IL-17, IL-36, and IL-20 families by comparative transcriptome and proteomics analysis using *Spink5* cKO mice and NS patients. Furthermore, they identified that KLK14 and KLK7 could modify the inflammation signaling via AHSG and IL-36 A and IL-36G processing, respectively. Finally, they analyzed the relationship between systemic inflammation and immune abnormalities in the thymus, lymph node, and spleen. Their findings could contribute to a better understanding of NS pathogenesis and the development of effective treatments. Overall, the methods used in this study were deemed appropriate, and the results and discussion were convincing in explaining the conclusions. Specific comments are shown below.

Major point

1. On page 4, line 142, the authors stated that "the severity of the skin lesion strongly correlated with the absence of LEKTI protein," supported by immunofluorescence images shown in Supplementary Fig2h. If there are additional quantitative data, such as Western blot results, please include them. And, why does it appear that the expression of LEKTI is decreased more in the lesional area than in the non-lesional area in *Spink5* cKO mice?

We agree with the reviewer's points.

We have revised the text in the manuscript that refers to this point and have performed additional measurements of *Spink5* expression at the mRNA and protein levels to better define lesional and non-lesional skin. *Spink5* cKO mice show mosaic somatic inactivation of *Spink5*, which is due to the incomplete floxed exon excision mediated by CreERT2. Lesional skin is a skin that has visible signs of skin barrier defect and inflammation such as scaling, crusts, redness, thickening and/or alopecia. These externally visible signs of lesional skin are also evident at the histology level (H&E staining of paraffin sections). On the contrary, non-lesional skin is outwardly normal-looking skin. At the histology level, however, the epidermis in non-lesional skin may be thicker than the epidermis of WT control skin.

We measured the expression of *Spink5* mRNA and LEKTI protein in lesional and non-lesional back skin samples from *Spink5* cKO mice ($n \geq 7$) and correlated their expression levels to the severity of skin lesions (Suppl. Fig.2i-o). In the initial submission, we correlated *Spink5* mRNA levels in lesional skin to skin severity score of the whole body (Suppl. Fig. 2i of the initial submission); however, samples for RNA or protein analyses are taken from mosaic *Spink5* mutant mice, and represent a small part of back skin lesions. Therefore, correlating the level of *Spink5*/LEKTI expression in these samples to whole-body skin severity score would not be correct. In this revised version of the manuscript, we corrected our approach and performed such correlation analyses using the *Spink5*/LEKTI expression level values and back skin lesion severity score, instead of whole-body skin score.

We found that the reduction of *Spink5* mRNA levels, as measured by RT-qPCR, correlates negatively and significantly with the severity of back skin lesions (Suppl. Fig.2n). The average level of *Spink5* mRNA detected in non-lesional skin was similar to that detected in back skin samples from WT control mice. On the contrary, in lesional skin, the average level of *Spink5* mRNA expression was reduced by half (Suppl. Fig. 2i).

We further analyzed the expression of LEKTI protein by immunofluorescence staining of *Spink5* cKO lesional and non-lesional back skin paraffin sections (Suppl. Fig. 2h and j) and by SDS-PAGE/Western blot of protein extracts from *Spink5* cKO lesional and non-lesional back skin samples in parallel with extracts from WT skin samples and *Spink5*^{-/-} skin samples (Suppl. Fig. 2k-m).

The LEKTI immunofluorescence images were quantified to calculate the ratio of LEKTI-positive epidermis area to total epidermis area. This ratio was significantly lower in lesional skin samples as compared to non-lesional skin ones (Suppl. Fig.2j). Moreover, there is a significant negative correlation between the ratio of LEKTI-positive area/total epidermis area and back skin lesion severity score (Suppl. Fig. 2o).

Finally, SDS-PAGE/Western blot analyses revealed reduction of full-length LEKTI (~120 kDa band) expression in lesional skin samples as compared to non-lesional skin and WT control skin samples (Suppl. Fig.2k-m). The antibody that was used for SDS-PAGE/Western blot is a polyclonal antibody that targets the C-terminal part of mouse LEKTI (Cys734~Ala995). LEKTI is known to be proteolytically cleaved into several fragments^{8,9}. We observed two specific bands of 25 kDa and 40 kDa that could correspond to LEKTI fragments (Suppl. Fig. 2k). There is only one previous study that shows SDS-PAGE/Western blot analyses of endogenous mouse LEKTI in extracts from newborn mouse skin using a different polyclonal antibody that also targets the C-terminal region of LEKTI⁹. In that study, the authors detected full-length LEKTI as well as two LEKTI fragments of 70-66 kDa. Further biochemical analyses are needed to confirm the identity of the 40-25 kDa bands in our analyses.

In conclusion, we confirmed a significant negative correlation between the severity of skin lesions and *Spink5*/LEKTI expression level, by providing more accurate measurement and comparisons at protein and mRNA levels as well as by adding additional animal numbers to the study. In addition, we provide an SDS-PAGE/Western blot quantification that confirms the significant decrease of LEKTI protein in lesional skin of *Spink5* cKO mice.

2. Was double staining for KRT14 and KRT10 conducted in Figure 2n? If the tissue section was cut at an angle, it is possible that the basal layer appears thicker, which could exaggerate the observed increase in KRT14 expression.

We performed double immunofluorescence staining of KRT14 and KRT10 in paraffin sections of *Spink5* cKO and WT control skin (Suppl. Fig. 4m-m'). We quantified co-localization of fluorescence signal from KRT14 (green) and KRT10 (red) in the epidermis by Pearson correlation analysis (Coloc 2 tool in Fiji). While KRT14 and KRT10 do not overlap in WT epidermis, the results from these measurements show that KRT14 and KRT10 staining signals partially overlap in *Spink5* cKO epidermis (positive correlation coefficient) (Suppl. Fig.4m').

3. On page 8, line 310, it is mentioned that CD34 is the marker of hair follicle stem cells. However, it is also one of the markers associated with fibroblasts. As a result, could the decrease in CD34-positive fibroblasts potentially lead to reducing collagen fibers?

We thank reviewer #2 for this suggestion.

We performed double immunofluorescence staining of the mesenchymal marker Vimentin (highly expressed in fibroblasts) and CD34 in paraffin sections of skin from *Spink5* cKO and control mice (Figure 4g). We performed a set of quantitative analyses of fluorescence microscopy images, where we quantified the number of Vimentin+ cells and the number of Vimentin+CD34+ cells (Figure 4h-j). The intensity threshold values that define Vimentin+ or CD34+ cells was initially set in WT control tissues, and then used to evaluate the number of Vimentin+ or/and CD34+ cells in *Spink5* cKO dermis.

We observed a slight, but not significant, increase (1.5-fold) of Vimentin+ cells in *Spink5* cKO dermis as compared to WT mice (Figure 4h). As expected, the expression of CD34 in *Spink5* cKO dermis was strongly reduced. Additionally, we observed that 46% of the Vim+ cells in WT dermis are also CD34+ (Figure 4j), while in *Spink5* cKO mouse skin the Vim+CD34+ sub-population was reduced to 4% from the total number of Vim+ cells. In addition, we measured the areas positively stained for collagens (blue color in paraffin sections stained with Masson's trichrome) using Fiji as previously described¹⁰. When compared to WT mouse skin paraffin sections stained with Masson's trichrome, we found a significantly lower blue stained area in *Spink5* cKO dermis (Figure 4f, Suppl. Fig. 8g).

Thus, as suggested by reviewer #2, it is possible that the reduction of collagen fibers that we see in *Spink5* cKO skin is due to the loss of CD34+ fibroblasts. Importantly, this reduction of collagen fibers in *Spink5* cKO skin is consistent with the decreased expression of several collagens as determined in the skin proteomics experiments (Figure 4e). We thank reviewer #2 for suggesting to explore CD34 as a marker of fibroblasts, as we believe that this additional set of experiments improve the manuscript.

Minor point

4. Have you ever checked if pepPG278 can alleviate skin inflammation in *Spink5* cKO mice? We have tested this inhibitor in a short pilot study by systemic administration in *Spink5* cKO mice, but because pepPG278 is not stable enough in mouse plasma (its half-life is about 4 hours), we could not see an effect. pepPG278 is currently being optimized to improve its stability in mouse plasma.

5. In Fig 3d, the white text with a yellow background in the center of the Venn diagram is hard to read. We have corrected this figure.

6. On page 11, line 420, which Figure shows the correlation between Spleen size and phenotype severity?

Page 11, line 420 refers to thymus size, not spleen size. Now, we have added a plot of correlation between thymus size and skin phenotype severity (Suppl. Fig.14k) and a plot of correlation between spleen size and skin phenotype severity (Suppl. Fig. 14h-i). The reduction of thymus size in *Spink5* cKO mice correlates significantly with whole body skin phenotype severity (Suppl. Fig.14k). The increase of spleen size in *Spink5* cKO mice does not correlate significantly with skin phenotype severity (Suppl. Fig. 14h-i).

7. Isn't Figure 7g correct for Figure 7f on page 12, line 489?

On page 12, line 489 of the initial manuscript (Figure 7f) should be (Figure 7h, Supplementary Table S4), which shows a heatmap of log2 fold change values of differentially expressed genes in *Spink5* cKO lymph nodes ordered according to similarity of biological process GO. In the current, revised version of the manuscript, we have changed the order of some panels in Figure 7, so that now the heatmap in Figure 7 is in panel f.

In addition, the fold change expression values of the genes shown on the heatmap can be consulted in Supplementary Table S4 (excel file).

Reviewer #3 (Remarks to the Author):

The manuscript describes a novel *Spink5* conditional knockout (cKO) mouse model for Netherton syndrome (NS), a severe and potentially lethal rare genetic skin disease. The interesting aspect of this model is the survival of the animals for several weeks, which should enable the investigation of new therapeutic approaches. This is a substantial improvement over previously described *Spink5* knockout models (perinatally lethal), skin grafting of neonatal *Spink5*^{-/-} skin onto normal mice (requiring the sacrifice of many animals) or knockout models of individual proteases (representing only partial aspects of the human disease, since intact LEKTI is still present).

The authors present an in-depth characterization of the model on the molecular, histologic, enzyme and functional levels and show a good correlation between the transcriptomes and proteomes of their knockout mice and the human disease. Therefore, the model will be of interest to other investigators in the field and may promote the identification of novel treatments.

The manuscript is a bit difficult to read in many places, because it is very long and the reader has to switch between so many parts of Figures and Supplementary Figures. It should be possible to reduce the information in the supplements or indicate the important information in individual gel graphs, e.g., by arrows. Some details about the comparison of *Spink5* cKO and Tg.hKLLK5 mice could also be shifted to the supplementary information or even, for the sake of clarity, be omitted from this paper, since the main goal is to show the validity of the new model for human NS.

The description of the experimental procedures is very detailed and should allow reproduction.

In the following, I will address individual aspects, which could be improved or should be changed.

Introduction:

1) The authors should reference Guo et al, 2022: Establishment of a mouse model of Netherton syndrome based on CRISPR/Cas9 technology, *Eur J Dermatol.* 2022 Jul 1;32(4):459-463. doi: 10.1684/ejd.2022.4287, and explain what differentiates their model from the one described by Guo et al.

We thank the reviewer's comment.

Now we have included this reference in the introduction. We would like to highlight that the mouse model described by Guo et al is a constitutive *Spink5*-deficient mouse model obtained by CRISPR/Cas9-mediated deletion of 22 bps in exon 3. These mice die in about 1 week after birth, which does not allow the study of disease progression and phenotype in adult stages in contrast to our viable *Spink5* cKO model.

Results:

2) Lines 118 – 125: Consider changing the order of spontaneous floxed allele excision \diamond induced deletion \diamond spontaneous floxed allele excision to describing first one and then the other.

We have changed the order.

3) Lines 141 – 145: While the lack of correlation between *Spink5* mRNA expression and skin severity score (Fig. S2i) is clear, Fig. S2h does not prove a correlation between the number of LEKTI-positive cells and phenotype severity, since only one mouse is shown.

We agree with the reviewer's point, thus we decided to extend as well as improve the accuracy of our measurements.

We measured the expression of *Spink5* mRNA and LEKTI protein in lesional and non-lesional back skin samples from *Spink5* cKO mice ($n \geq 7$) and correlated their expression levels to the severity of skin lesions (Suppl. Fig. 2i-o). In the initial submission, we correlated *Spink5* mRNA levels in lesional skin to skin severity score of the whole body (Suppl. Fig. 2i of the initial submission); however, samples for RNA or protein analyses are taken from mosaic *Spink5* mutant mice, and represent a small part of back skin lesions. Therefore, correlating the level of *Spink5*/LEKTI expression in these samples to whole-body skin severity score would not be correct. In this revised version of the manuscript, we corrected our approach and performed such correlation analyses using the *Spink5*/LEKTI expression level values and back skin lesion severity score, instead of whole-body skin score.

The average level of *Spink5* mRNA detected in non-lesional skin was similar to the level detected in back skin samples from WT control mice. In turn, the average level of *Spink5* mRNA expression was reduced by half in lesional skin (Suppl. Fig. 2i). We found that the reduction of *Spink5* mRNA levels correlated negatively and significantly with the severity of back skin lesions (Suppl. Fig. 2n).

We further analyzed the expression of LEKTI protein by immunofluorescence staining of *Spink5* cKO lesional and non-lesional back skin paraffin sections (Suppl. Fig. 2h and j). The LEKTI immunofluorescence images were quantified to calculate the ratio of LEKTI-positive epidermis area to total epidermis area. This ratio was significantly lower in lesional skin samples as compared to non-lesional skin ones (Suppl. Fig. 2j). Moreover, there is a significant negative correlation between the ratio of LEKTI-positive area/total epidermis area and back skin lesion severity score (Suppl. Fig. 2o).

4) Line 161: "(Figure 2b)" should read (Figure 2a).

We have corrected this.

5) Line 181: should read ... Supplementary Figure S4d-e.

We have corrected this.

6) Line 193: should read ...Supplementary Figure S4h-k'.

We have corrected this.

7) Lines 193 – 196: I cannot follow the author's argumentation that the overall expression of FLG, DSG1 and DSP were unchanged. According to Suppl. Fig. S4j', DSG1 integrated fluorescence intensity is even increased, as I would assume from Fig. 2i. This is rather surprising, considering that DSG1 is a KLK substrate and reduced in NS skin. The authors should comment on the increased staining of FLG, DSG1 and DSP in Figures 2h – j.

We understand the reviewer's concerns. Accordingly, we have now included quantification of the DSP immunofluorescence staining images (Suppl. Fig. 4l-l').

By visual inspection of the immunofluorescence staining images, there seem to be higher levels of DSG1, FLG and DSP in *Spink5* cKO skin than in WT control skin. However, quantification of mean fluorescence intensity of the staining signal in the epidermis (the sum of all pixel intensities in the epidermis divided by the number of pixels in the epidermis, i.e. intensity per unit area) does not show

significant decreases in the protein levels of DSG1, FLG or DSP in *Spink5* cKO skin vs WT control skin (Fig. 2h-j, Suppl. Fig. 4i-j', l-l'). These results are in agreement with the skin transcriptomics and skin proteomics data (Figure R1). *Dsg1* and *Dsp* are not differentially expressed in *Spink5* cKO skin and NS patient skin (skin transcriptomics, Figure R1a-b). *Flg* is also not differentially expressed in NS patient lesional skin (Figure R1b), but in *Spink5* cKO skin *Flg* is differentially up-regulated at the mRNA level (Figure R1a). The protein levels of DSG1, DSP and FLG are unchanged in *Spink5* cKO and NS patient lesional skin as compared to WT/healthy control skin (skin proteomics dataset, Figure R1c-d).

By contrast, quantification of integrated fluorescence intensity (the sum of all pixel intensities in the epidermis), shows an increase of protein levels in *Spink5* cKO epidermis, which are statistically significant for DSG1 (Suppl. Fig. 4i', j' and l'). This difference, as compared to the mean fluorescence intensities measured for these three markers, is caused by the hyperplastic epidermis in *Spink5* cKO skin. Therefore, the visual impression that DSG1, FLG and DSP protein levels are higher in the immunofluorescence staining images (Fig.2h-j) might derive from the fact that the epidermis in *Spink5* cKO mice is thicker and thus the area, where these markers are expressed, is larger as compared to WT skin.

Figure R1. Comparison of RNA and protein expression levels of skin barrier genes in *Spink5* cKO mice and NS patients' lesional skin.

(a-b) Bar plots showing the mRNA expression level of genes coding for structural components of the stratum corneum and the granular layers of the epidermis in *Spink5* cKO (a) and NS patients' (b) lesional skin. Log₂ fold change values from RNAseq data analyses are plotted.

(c-d) Bar plots showing expression level of stratum corneum and granular layer structural proteins in *Spink5* cKO **(c)** and NS patients' **(d)** lesional skin. Log₂ fold change values from LC-MS/MS proteomics data analyses are plotted.

In **(a-d)**, columns in red or blue correspond to genes/proteins that are significantly (adjusted *P*-value < 0.05) up-regulated or down-regulated, respectively. Black columns correspond to genes/proteins, whose expression levels are not significantly changed. In **(c-d)**, asterisk next to the protein symbol indicate that the corresponding protein was not detected with the LC-MS/MS proteomics method used.

Additionally, here we provide reviewer #3 with further data and reinforcing arguments: DSG1 is a substrate of KLK5¹¹ and its protein levels are expected to be lower in *Spink5* cKO and NS patient skin. The levels of mouse DSG1 were previously shown to be diminished in the skin of *Spink5*^{-/-} mice¹². However, the levels of DSG1 were not diminished in another constitutive *Spink5*-deficient model¹³. The protein levels of DSG1 in NS patient skin were previously measured using immunohistochemical staining in skin sections, showing that DSG1 is reduced or absent only in the upper granular layers of the epidermis, but is still expressed in the lower spinous layer¹⁴. Since the epidermis in NS patient skin is thicker, the expression of DSG1 in the lower spinous layers could compensate for the loss of DSG1 in the superficial layer with no net loss of DSG1 in the epidermis, which would explain the skin mass spectrometry findings (Figure R1d).

KLK5 is also involved in the proteolytic processing of pro-FLG to FLG monomers^{12,15,16}. The anti-FLG antibody used in our study is a monospecific polyclonal antibody raised against a peptide corresponding to residues 24-39 in the partial sequence of filaggrin, thus it is expected to detect mostly FLG monomers. Detection of FLG by mass spectrometry would not distinguish between pro-FLG and FLG monomer peptides.

8) Line 283: "... which is driven by IL-17, IL-36 and IL-20 family members" is formulated a bit too strongly. Just because the mRNAs for these cytokines are upregulated does not necessarily imply a causative role for the disease process.

We agree with the reviewer's point. We have changed this to "..., which is characterized by IL-17, IL-36 and IL-20 family cytokine ..."

9) Lines 339 – 341: it would be sufficient to show either Fig. 5e or Fig 5f (5f preferred for the sake of comparison with 5g and 5h).

We have removed figure 5e.

10) Lines 341 and 343: Suppl. Fig. 12 does not contain information about AHSG and should, therefore, be quoted differently.

In the original manuscript, supplementary Figure 12 contains information about the activity of the enzymes used in the in vitro digestion assays for AHSG, IL-36A and IL-36G that were described in this section. We therefore think that referring to this figure at this stage is pertinent.

11) Line 357: Suppl. Fig. 11l does not provide any information about cleavage of IL-36A, in fact, it is redundant with S11d-k and could be left out.

In the original manuscript, supplementary Figure 11l shows negative control digestion reactions (only enzyme, or only substrate) for KLK5 and TMPRSS11D *in vitro* cleavage assays shown in Supplementary Figures 11f,g,i and k.

12) Lines 357 – 359: “IL-36G, another member of the IL-36 pro-inflammatory cytokine family that is highly upregulated at the mRNA and protein levels in skin of NS patients...”. This is in direct contrast to line 351, “IL-36A protein could not be detected in NS patients...”. Please clarify!

We would like to clarify a possible misunderstanding. IL-36A and IL-36G are both members of the IL-36 cytokine subfamily, but have distinct regulation and pro-inflammatory functions. IL-36G is up-regulated at the mRNA and protein levels in both *Spink5* cKO and NS patient skin. IL-36A is strongly up-regulated at the mRNA level in both *Spink5* cKO and NS patient skin, but its protein levels are much lower than the expected. In fact, we could not detect IL-36A in immunofluorescence staining of NS patient skin biopsies nor in the mass spectrometry analyses of NS patient skin biopsies. In *Spink5* cKO skin, we do detect increased IL-36A levels relative to WT skin, but this increase is much lower than the expected given the high expression at the mRNA level. IL-36A was found in our skin transcriptome-proteome comparative analyses (Figure 5b and 5d) as a gene whose expression level could be regulated at the posttranslational level, i.e. by proteolytic degradation. We then show that rhIL-36A can be degraded by rhKLK14 (Figure 5f, Supplementary Figure 11d), thereby supporting the hypothesis of a possible posttranslational regulation of IL-36A in *Spink5* cKO and NS patient skin.

Since both IL-36A and IL-36G are part of the same cytokine family with important pro-inflammatory role in the skin and we found that KLKs can induce an *in vitro* cleavage IL-36A, we decided to extend our *in vitro* cleavage analyses to IL-36G.

13) Line 365: “...proteomics analyses confirm IL-36 and IL-17 signaling...”. Since IL-17 has not been mentioned in the proteomics results section so far, this statement comes rather as a surprise. Please explain!

In the initial submission, we mentioned IL-17 in the proteomics results section (see line 307 of the initial submission manuscript).

IL-17 and IL-36 signaling pathways have overlapping downstream target genes and both signaling pathways converge on neutrophil recruitment^{17,18}. In our skin proteomics data, we identified increased levels of neutrophil-related proteins such as S100A8/9, LTF, ARG1 or MPO. Therefore, we think that this statement is pertinent.

14) Lines 382 – 383: the distribution of graphs between Fig. 6 and S13 is difficult to understand. Why are some markers quantified in Fig 6, but not others, which would be equally important (e.g., Fig. 6g-k, Fig. S13h)?

We have now provided quantification of the markers shown in Figure 6g-k and Supplementary Figure 13h. See Figure 6f'-k' and Suppl. Figure 13g-j''.

15) Line 391: IL-36A ... was strongly upregulated in epidermal keratinocytes: how does this fit in with IL-36A being degraded by kallikreins?

We found that IL-36A is strongly up-regulated at the mRNA level in both *Spink5* cKO and NS patient skin, but its protein levels are much lower than the expected. In *Spink5* cKO skin, we do detect increased IL-36A levels relative to WT skin, but this increase is much lower than the expected given the high expression at the mRNA level. IL-36A was found in our skin transcriptome-proteome comparative analyses (Figure 5b and 5d) as a gene, whose expression level could be regulated at the

posttranslational level, i.e. by proteolytic degradation, which we tested using in vitro digestion assays. IL-36A protein levels in the skin of *Spink5* cKO mice can still be higher as compared to WT mice, despite possible cleavage/degradation by epidermal KLKs.

Moreover, quantification of staining intensity signal in the immunofluorescence images in Figure 6g-k and Supplementary Figure 13h revealed that the protein levels of IL-36A, IL-24, S100A9 and S100A8 in the epidermis of *Spink5* cKO mice are significantly higher as compared to WT control mice. The fold increase values of signal in *Spink5* cKO epidermis vs WT epidermis were as follows:

IL-36A: mean intensity = 4, integrated intensity = 46

IL-24: mean intensity = 15, integrated intensity = 116

S100A9: mean intensity = 25, integrated intensity = 157

S100A8: mean intensity = 24.5, integrated intensity = 195

Because the protein expression of IL-36A in *Spink5* cKO skin shows lower fold increase as compared to IL-24, S100A9 and S100A8, we have modified the text in lines 390 – 394 as follows:

The pro-inflammatory cytokine IL-36A was up-regulated in the epidermis of *Spink5* cKO mice (4-fold increase), showing highest expression in the upper epidermal layers (Figure 6g, g'). The expression of IL-24 was strongly increased (15-fold) in all epidermal layers of *Spink5* cKO mice (Figure 6h, h').

16) Line 420: Fig. S14c and d do not show a correlation between thymus size and severity.

We have now added a plot showing a correlation between thymus size and whole-body skin phenotype severity in *Spink5* cKO mice (Suppl. Fig. 14k).

17) Lines 422-423: Fig. 7f-g seem to be identical to Suppl. Fig. S15a and b, except for the label of the y-axis. Please clarify!

The flow cytometry data is presented as both cell counts (Figure 7f-g of the initial manuscript, Figure 7g-h of the current revised manuscript) and frequency (Suppl. Fig. S15a-b of the initial manuscript, Suppl. Fig. 17a-b of the current revised manuscript). The cell counts (Figure 7g-h of the current revised manuscript) for each population were calculated by multiplying the number of viable cells in the tissue sample (as determined by Trypan Blue exclusion assay) by the frequency (Suppl. Fig. 17a-b of the current revised manuscript) of the cell population as determined by flow cytometry. For the thymocyte cell subsets, we obtain similar results when the data is presented as cell counts and when it is presented as frequencies. We decided to also show the frequency data, since it can provide information on whether a cell population has expanded or shrunk as compared to the other cell populations.

18) Line 427: The SP thymocytes are shown in Fig. 7g, not 7f.

We have corrected this. In the current, revised manuscript the SP thymocytes are shown in Figure 7h.

19) Line 464: please explain how the *Spink5* cKO mice could have been infected with *S. aureus*, since they were kept in a specific pathogen-free facility in IVC cages. Where does the infection come in?

Mice housed in a specific pathogen-free (SPF) facility are not germ-free. SPF mice in our facility are free of specific pathogens monitored by routine testing, but they can harbor opportunistic pathogens

such as *S.aureus*, for which routine surveillance is not performed. *Spink5* cKO mice could have been infected with *S.aureus* during manipulation of cages/mice by the staff.

20) Lines 488-489: “Fig 7f” should read S18f. The fold change of 67 cannot be deduced from this graph.

On page 12, line 489 of the initial manuscript (Figure 7f) should be (Figure 7h, Supplementary Table S4), which shows a heatmap of log₂ fold change values of differentially expressed genes in *Spink5* cKO lymph nodes ordered according to similarity of biological process GO. In the current, revised version of the manuscript, we have changed the order of some panels in Figure 7, so that now the heatmap in Figure 7 is in panel f.

In addition, the fold change expression values of the genes shown on the heatmap can be consulted in Supplementary Table S4 (excel file).

21) Lines 490 – 494: It is not clear what data they relate to.

In the initial submission, lines 490 – 494 relate to Figure 7h, Supplementary Figure S18a-b, and Supplementary Table 4. We have now added this reference at the end of the paragraph (line 470 and line 472) with the revised numbering of the figures as follows: Figure 7f and Suppl. Table 4.

Discussion:

The discussion is comprehensive, well balanced and presents points for further investigation.

Materials and Methods:

This section is very detailed and should allow other investigators to replicate the results.

22) Lines 679 – 682: The authors may want to explain why they used either i.p.- or topical induction of the knockout phenotype and whether there were any differences in outcome.

We did not see differences in the outcome between topical and i.p. tamoxifen-mediated induction of CreERT2. Topical induction of CreERT2 was preferred since it is non-invasive.

We found that CreERT2-mediated *Spink5* floxed exon excision and subsequent development of NS-like skin phenotype is more efficient in mice that already show some basal level of spontaneous CreERT2 activity (assessed by the presence of the excised allele). Thus, we routinely screened all mice for spontaneous, tamoxifen-independent floxed allele excision.

In mice with mild signs of phenotype due to tamoxifen-independent CreERT2 activation (e.g. only visible as slight scaling on the ears or on the muzzle), shorter induction (with 2-3 applications of topical 4-hydroxytamoxifen) was sufficient to induce whole body knock-out phenotype. In such mice, topical 4-hydroxytamoxifen applied on the back skin could also induce CreERT2 activity in the skin of the belly, face, or extremities and in internal organs where the KRT14 promoter is active, such as esophagus, tongue or thymus.

We have modified the Material and Methods section accordingly.

Figure Legends:

23) Line 1545: If the organ weight index were calculated as described, the values should be <1. Please replace by the explanation in line 1900.

We have corrected this.

24) Lines 1643 – 1644: Instead of saying “...from a *Spink5* cKO mouse”, the authors may want to point out that this is the mouse of Fig. 2d.

We have added this information.

25) Lines 1782 – 1784: It would be helpful for the reader to mention not only Fig. S10, but also Fig. S9d-e.

We have added this information.

Figures:

26) Figure 5: Fig. 5d could be moved to the supplements in order to focus more on the comparison *Spink5* cKO to human NS in the main part of the document. Fig. 5e may not be necessary (cf. above).

We have removed Figure 5e.

We prefer to keep Figure 5d in the main figure. Figure 5d shows the results from the skin transcriptome-proteome comparative analyses in *Spink5* cKO and *Tg.hKLK5* mice, where we identified IL-36A as a possible target of posttranslational modification. Thus, Figure 5d is directly linked to Figures 5g and e of the initial manuscript. We strongly believe that if we move Figure 5d to the supplementary data, it will be more difficult for the reader to follow the manuscript. The skin transcriptome-proteome comparative analyses in *Spink5* cKO and *Tg.hKLK5* mice were done to enlarge the pool of possible candidates for posttranslational modification, since comparing proteomes between two mouse models results in more mRNA-protein pairs (intra-species comparison) as opposed to comparing mouse and human proteomes, where there is a lower number of shared homologous mRNA-protein pairs.

27) Suppl. Fig. 17a-d: What does the abbreviation dLN stand for? The figure legend mentions inguinal lymph nodes.

dLN stands for draining lymph node. For clarity, we have replaced this abbreviation by “lymph node”.

Finally, we would like to thank reviewer #3 for the critical reading of our manuscript, for the detailed review of all sections, the corrections and the suggestions for improvement.

References:

1. Brennecke, P. *et al.* Single-cell transcriptome analysis reveals coordinated ectopic gene-expression patterns in medullary thymic epithelial cells. *Nat. Immunol.* **16**, 933–941 (2015).
2. Remot, A. *et al.* Bacteria isolated from lung modulate asthma susceptibility in mice. *ISME J.* **11**, 1061–1074 (2017).
3. Leinwand, J. C. *et al.* Intrahepatic microbes govern liver immunity by programming NKT cells. *J. Clin. Invest.* **132**, (2022).
4. Acosta, E. M. *et al.* Bacterial DNA on the skin surface overrepresents the viable skin

microbiome. *Elife* **12**, (2023).

5. Oikonomopoulou, K. *et al.* Induction of complement C3a receptor responses by kallikrein-related peptidase 14. *J. Immunol.* **191**, 3858–3866 (2013).
6. Nylander-Lundqvist, E. & Egelrud, T. Formation of active IL-1 beta from pro-IL-1 beta catalyzed by stratum corneum chymotryptic enzyme in vitro. *Acta Derm Venereol.* **77**, 203–209 (1997).
7. Yamasaki, K. *et al.* Kallikrein-mediated proteolysis regulates the antimicrobial effects of cathelicidins in skin. *FASEB J. Off. Publ. Fed. Am. Soc. Exp. Biol.* **20**, 2068–2080 (2006).
8. Fortugno, P. *et al.* Proteolytic activation cascade of the Netherton syndrome-defective protein, LEKTI, in the epidermis: implications for skin homeostasis. *J. Invest. Dermatol.* **131**, 2223–2232 (2011).
9. Galliano, M. F. *et al.* Characterization and expression analysis of the Spink5 gene, the mouse ortholog of the defective gene in Netherton syndrome. *Genomics* **85**, 483–492 (2005).
10. Chen, Y., Yu, Q. & Xu, C. A convenient method for quantifying collagen fibers in atherosclerotic lesions by ImageJ software. in (2017).
11. Caubet, C. *et al.* Degradation of corneodesmosome proteins by two serine proteases of the kallikrein family, SCTE/KLK5/hK5 and SCCE/KLK7/hK7. *J. Invest. Dermatol.* **122**, 1235–1244 (2004).
12. Descargues, P. *et al.* Spink5-deficient mice mimic Netherton syndrome through degradation of desmoglein 1 by epidermal protease hyperactivity. *Nat. Genet.* **37**, 56–65 (2005).
13. Yang, T. *et al.* Epidermal detachment, desmosomal dissociation, and destabilization of corneodesmosin in Spink5^{-/-} mice. *Genes Dev.* **18**, 2354–2358 (2004).
14. Descargues, P. *et al.* Corneodesmosomal cadherins are preferential targets of stratum corneum trypsin- and chymotrypsin-like hyperactivity in Netherton syndrome. *J. Invest. Dermatol.* **126**, 1622–1632 (2006).
15. Hewett, D. R. *et al.* Lethal, neonatal ichthyosis with increased proteolytic processing of filaggrin in a mouse model of Netherton syndrome. *Hum. Mol. Genet.* **14**, 335–346 (2005).
16. Sakabe, J. *et al.* Kallikrein-related peptidase 5 functions in proteolytic processing of profilaggrin in cultured human keratinocytes. *J. Biol. Chem.* **288**, 17179–17189 (2013).
17. Carrier, Y. *et al.* Inter-regulation of Th17 cytokines and the IL-36 cytokines in vitro and in vivo: implications in psoriasis pathogenesis. *J. Invest. Dermatol.* **131**, 2428–2437 (2011).
18. Pfaff, C. M., Marquardt, Y., Fietkau, K., Baron, J. M. & Lüscher, B. The psoriasis-associated IL-17A induces and cooperates with IL-36 cytokines to control keratinocyte differentiation and function. *Sci. Rep.* **7**, 15631 (2017).

REVIEWERS' COMMENTS:

Reviewer #1 (Remarks to the Author):

The authors have sufficiently addressed the questions I have. Congratulations on the great work!

Reviewer #2 (Remarks to the Author):

The revised manuscript is outstanding. The authors have continued to enhance the quality and interpretation of the data in light of the reviewers' suggestions. I have no further comments.

Reviewer #3 (Remarks to the Author):

Reviewer 3 thanks the authors for carefully addressing all the points in a satisfactory fashion.

REVIEWERS' COMMENTS:

Reviewer #1 (Remarks to the Author):

The authors have sufficiently addressed the questions I have. Congratulations on the great work!

Reviewer #2 (Remarks to the Author):

The revised manuscript is outstanding. The authors have continued to enhance the quality and interpretation of the data in light of the reviewers' suggestions. I have no further comments.

Reviewer #3 (Remarks to the Author):

Reviewer 3 thanks the authors for carefully addressing all the points in a satisfactory fashion.

We thank all reviewers for reading our revised manuscript. We thank all reviewers once again for the initial review of our manuscript and for providing invaluable feedback and suggestions that helped us improve the manuscript.